# A Novel General Framework for Sharp Lower Bounds in Succinct Stochastic Bandits

**Guo Zeng**
School of Computing and Information Systems, The University of Melbourne
guo.zeng@student.unimelb.edu.au

**Jean Honorio**
School of Computing and Information Systems, The University of Melbourne, and
ARC Training Centre in Optimisation Technologies, Integrated Methodologies, and Applications
jean.honorio@unimelb.edu.au

## Abstract

Many online learning applications adopt the stochastic bandit problem with a linear reward model, where the unknown parameter exhibits a succinct structure. We study minimax regret lower bounds which allow us to know whether more efficient algorithms can be proposed. We introduce a general definition of succinctness and propose a novel framework for constructing minimax regret lower bounds based on an information-regret trade-off. When applied to entry-sparse vector, our framework sharpens a recent lower bound by [7]. We further apply our framework to derive novel results. To the best of our knowledge, we provide the first lower bounds for the group-sparse and low-rank matrix settings.

## 1 Introduction

Stochastic bandits [4] model online learning problems, where an agent is evaluated over its interactions with an initially unknown and memory-less bandit machine. At each round $t$, the agent (stochastically) plays an action $A_t$ to the machine and (stochastically) receives a real-valued reward $y_t$. The memory-less nature allows defining an optimal action that maximizes $\mathbb{E}[y_t]$, independent of the past interactions. Against those optimal actions, the agent's regret up to $n$ rounds on a specific bandit instance can be calculated. Assuming bounded single-round regret, it is desirable for algorithms to achieve sub-linear regret rate $o(n)$ across a class of bandit instances, called the assumption class.

In real-world applications like advertising recommendation [6] and personalized healthcare systems [3], automated decision-making often involves high-dimensional data. To enable effective algorithm design, one common modeling approach assumes a linear reward: $\mathbb{E}[y_t] = \langle A_t, \theta \rangle$, where the action is represented by an associated data vector or matrix $A_t$, and the bandit instance by a parameter $\theta$ of the same shape but with a succinct structure. For example, entry-sparse bandits—also known as Lasso bandits or sparse linear bandits—assume that $\theta \in \mathbb{R}^d$ has its number of non-zero entries limited to $s \ll d$ [5]. Alternatively, if $\theta$ is organized into a matrix in $\mathbb{R}^{d_1 \times d_2}$, it may be constrained to have a low rank $s \ll \min(d_1, d_2)$, to be group-sparse with no more than $s \ll d_1$ non-zero rows, or to exhibit other structured forms [12]. We consider these bandits to be succinct linear bandits, differing in how they impose succinct constraints on $\theta$.

While upper bounds guarantee the regret growth of a specific algorithm across an assumption class, minimax lower bounds apply to all possible algorithms on that assumption class, and are usually established by identifying hard bandit instances tailored to any given algorithm. To illustrate, suppose that $A_t \in \mathbb{R}^d$ is to be chosen from the standard basis $\{e_1, e_2, ..., e_d\}$ and $\theta \in \mathbb{R}^d$ has only one non-zero entry. When $d > n$, given any algorithm, there obviously exists a tailored $\theta$ to cause that

Table 1: Some results obtained with our novel framework, compared to existing upper bounds from algorithms and prior lower bounds. The number of rounds is denoted by $n$. For entry-sparse vectors, $d$ is the dimension and $s$ is the number of non-zero entries. For low-rank matrices and group-sparse matrices, $d_1$ is the number of rows and $d_2$ is the number of columns. For low-rank matrices, $s$ is the matrix rank, while for group-sparse matrices, $s$ is the number of non-zero rows. $C_{\min}$ is a quantity measuring to what extent the action set admits a well-condition exploration, properly introduced in Definition 4.1.

| | Upper Bound from Algorithms | Prior Lower Bound | Our Lower Bound |
|---|---|---|---|
| Entry-sparse vector | $O(\sqrt{sdn})$ in [1] $O(s\sqrt{n}\log(dn))$ in [10] $O(\sqrt{sn\log(dn)})$ in [13] $O(C_{\min}^{-2/3}s^{2/3}n^{2/3})$ in [7] $O(s^{1/3}n^{2/3}\sqrt{\log(dn)})$ in [12] | $\Omega(\min(C_{\min}^{-1/3}s^{1/3}n^{2/3}, \sqrt{dn}))$ in [7] $\Omega(\sqrt{sdn})$ in [11] | $\Omega(\min(C_{\min}^{-1/3}s^{2/3}n^{2/3}, \sqrt{dsn}))$ in Corollary 4.1 |
| Low-rank matrix | $O((d_1+d_2)^{3/2}\sqrt{sn})$ in [9] $O((d_1+d_2)^{3/2}\sqrt{sn})$ in [8] $O(s^{1/3}n^{2/3}\log(d_1+d_2))$ in [12] | None | $\Omega(\min(C_{\min}^{-1/3}s^{2/3}n^{2/3}, \sqrt{d_1d_2sn}))$ in Corollary 4.3 |
| Group-sparse matrix | $O(\sqrt{sd_2d_1n})$ in [8] $O(s^{1/3}n^{2/3}(\sqrt{d_2}+\sqrt{\log d_1}))$ in [12] | None | $\Omega(\min(C_{\min}^{-1/3}s^{1/3}n^{2/3}, \sqrt{d_1d_2n}))$ in Corollary 4.2 |

algorithm linear regret. Since the entry-sparse $\theta$ with $s = 1$ can also be interpreted as low-rank or group-sparse, the $\Omega(n)$ regret is unavoidable in the data-poor regime ($n \ll d$) for succinct linear bandits without extra constraints.

**Related Work.** For entry-sparse vector bandits, [11] gives a lower bound $\Omega(\sqrt{sdn})$ in the data-rich regime by emulating in parallel $s$ canonical multi-armed bandits, each with $d/s$ arms and a lower bound $\Omega(\sqrt{(d/s)n})$ established by [2]. A matching upper bound $O(\sqrt{sdn})$ is achieved by an online-to-confidence-set conversion algorithm in [1], which works on arbitrary action set and thus confirm the optimal rate $\Theta(\sqrt{sdn})$ in the data-rich regime. To ease the polynomial dependence on $d$, various constraints are introduced to study refined assumption classes [1, 10, 7, 13, 14], most of them focusing on upper bound analysis for their proposed algorithms. One closely related previous study by [7] considers entry-sparse bandits with a fixed action set $\mathcal{A}$ that allows for a well-explored sampling distribution, where the minimum eigenvalue of the population covariance matrix can be lower bounded despite growing $d$. By creating an information-regret trade-off, they construct a lower bound of $\Omega\big(\min(C_{\min}^{-1/3}(\mathcal{A})s^{1/3}n^{2/3}, \sqrt{dn})\big)$, where $C_{\min}(\mathcal{A})$ is the lower-bounding quantity. This lower bound nearly matches the upper bound $O(C_{\min}^{-2/3}(\mathcal{A})s^{2/3}n^{2/3})$ for their explore-sparsity-then-commit algorithm in the data-poor regime. It is also popular to study the contextual setting [10, 13, 14], where action $A_t$ is chosen from a set of stochastically generated contexts, making it easy to impose complicated assumptions on the more passively available action data $(A_t)_{t=1}^n$.

Other succinct linear bandits are less investigated. For low-rank matrix bandits, an upper bound of $O((d_1+d)^{3/2}\sqrt{sn})$ is established for different approaches: [9] developed an explore-subspace-then-refine algorithm for a bi-linear bandit setting, where the reward is computed as the bi-linear product of the matrix parameter with the left and right arms. [8] extended the algorithm to the generalized linear setting under a mild assumption on action data and the singular values of the low-rank parameter. For group-sparse matrix bandits, $O(\sqrt{sd_1d_2n})$ is given by [8] in an attempt to unify bandits with structured parameter in the contextual setting by constructing high-probability confidence ellipsoids. Under the restricted eigenvalue condition in the contextual setting, [12] offers a general framework of explore-sparsity-and-commit algorithms to achieve a common upper bound of $O(s^{1/3}n^{2/3})$ on entry-sparse, low-rank, group-sparse bandits, and their novel formulation of multi-agent bandits.

Overall, lower bound analysis for succinct linear bandits has been relatively under-developed, especially beyond the entry-sparse setting, where existing work often relies on informal arguments or naive reductions to the entry-sparse case to invoke known lower bounds. Meanwhile, upper bounds with dominating dependency on $d$ are proven attainable primarily in the data-rich regime [1, 10],

while dimension-free rates are typically achieved by constraining the assumption class to bandit instances that permit well-explored action data $(A_t)_{t=1}^n$ in order to bypass the linear lower bound in the data-poor regime [7, 12].

**Contributions.** Our contributions in this work are threefold. First, in Section 3.1, we propose a succinctness model in general vector space, along with lemmas that may be of independent interest. Second, in Section 3.2, we develop a general framework for deriving minimax lower bounds of succinct linear bandits in both data-rich and data-poor regimes, offering customizable constructions that revolve around two concepts, i.e. information-regret trade-off and succinctness support. Third, in Section 4, we apply this framework to three stochastic linear bandit problems that exhibit succinct structure and permit well-conditioned exploration. With minimal customization, we are able to improve the previous bound of [7] for entry-sparse vector bandits. To the best of our knowledge, we obtain the first lower bounds for group-sparse matrix bandits as well as for low-rank matrix bandits. These results, summarized in Table 1, showcase the generality of our framework and shed light to directions for extending existing upper and lower bounds.

## 2 Preliminaries

In this section, we start with a concise description of vector spaces, stochastic linear bandits and the notation used throughout this paper. We then briefly review the standard machinery commonly used in the literature for lower bound construction—techniques that we also draw upon later in proving our main theorem.

### 2.1 Problem setting

**Vector Space.** We consider a vector space $\mathbb{V}$ over $\mathbb{R}$, which is a set equipped with the vector addition and the scalar multiplication operations satisfying the standard axioms, such as the existence of $\vec{0}$ and additive inverse $-X$ of any vector $X \in \mathbb{V}$. We also define an inner product $\langle \cdot, \cdot \rangle$ as a function $\mathbb{V} \times \mathbb{V} \to \mathbb{R}$ that satisfies the properties of symmetry, linearity, and positive definiteness. Later, we will use the general definition of a norm, e.g. $P(\cdot) : \mathbb{V} \to \mathbb{R}$, which satisfies the triangle inequality $P(X + Y) \leq P(X) + P(Y)$, absolute scalability $P(aX) = |a|P(X)$, and positive definiteness $X \neq \vec{0} \Rightarrow P(X) > 0$, for all vectors $X, Y \in \mathbb{V}$ and scalars $a, b \in \mathbb{R}$.

**Stochastic Linear Bandit.** At each round $t$, the agent perceives a fixed action set $\mathcal{A} \subseteq \mathbb{V}$ and executes an action $A_t \in \mathcal{A}$. Then, the bandit machine generates a reward $y_t = \langle A_t, \theta \rangle + \eta_t$, where $\eta_t \sim \mathcal{N}(0, 1)$ and some parameter $\theta \in \mathbb{V}$ characterizes a bandit instance. The agent is conceptualized to follow a policy $\pi$ which yields $A_t$ stochastically conditioned on the ongoing interaction $A_1 y_1 A_2 y_2 \dots A_{t-1} y_{t-1}$.

Now that a stochastic process can be fully determined from the interactions between the policy $\pi$ and the bandit instance $(\mathcal{A}, \theta)$, or briefly $\theta$, up to a horizon of $n$ rounds, we can calculate the regret of $\pi$, compared to an omniscient policy that always plays an optimal action:

$$R_n^\pi(\mathcal{A}, \theta) := \mathbb{E}_{\pi,\theta} \left[ \sum_{t=1}^n \max_{X \in \mathcal{A}} \langle X, \theta \rangle - \sum_{t=1}^n y_t \right] \quad \text{where } \max_{X \in \mathcal{A}} |\langle X, \theta \rangle| \leq 1 \tag{1}$$

**Notation.** We denote the sub-optimal gap of an action $X \in \mathcal{A}$ under the instance $\theta$ as $\Delta_\theta(X) := \max_{X^* \in \mathcal{A}} \langle X^*, \theta \rangle - \langle X, \theta \rangle$. Given an inner product, we define the induced norm or length $\|X\| := \sqrt{\langle X, X \rangle}$ for all $X \in \mathbb{V}$, which can be shown to be a norm. We shorthand the set of integers $\{1, 2, 3, \dots, d\}$ as $[d]$, and $\{2, 3, \dots, d\}$ as $[2 : d]$. Let $|\mathcal{G}|$ be the cardinality of a set $\mathcal{G}$. The indicator $\mathbf{1}\{D\}$ equals 1 if the event $D$ occurs and 0 otherwise. We will omit $\pi$ in $\mathbb{P}_{\pi,\theta}$ and $\mathbb{E}_{\pi,\theta}$ when the context is clear. For vectors in $\mathbb{R}^d$, we use $\|\cdot\|_\infty$ and $\|\cdot\|_1$ to denote the $l_\infty$ and the $l_1$ norms. For matrices $X \in \mathbb{R}^{d_1 \times d_2}$, $\|X\|_{\infty,2} := \max_i (\sum_j x_{ij}^2)^{1/2}$ and $\|X\|_{1,2} := \sum_i (\sum_j x_{ij}^2)^{1/2}$ denote the $l_{\infty,2}$ and the $l_{1,2}$ norms, while $\|\cdot\|_{\text{op}}$ and $\|\cdot\|_{\text{nuc}}$ denote the operator and the nuclear norms.

## 2.2 Techniques for lower bound construction

**Le Cam's Lemma.** The construction of minimax lower bounds can be viewed as mirroring Le Cam's method in statistics for lower-bounding the worst-case error for any estimator. Let $\mathcal{F}$ be a set of distributions, where each distribution $\mathbb{P} \in \mathcal{F}$ is associated with a parameter $\theta \in \Theta$. Consider $\hat{\theta}(S)$ as an estimator for $\theta$ using empirical data $S \sim \mathbb{P}$. Let metric $d : \Theta \times \Theta \to [0, \infty)$ satisfy symmetry and triangle inequality. For any $\mathbb{P}_1, \mathbb{P}_2 \in \mathcal{F}$ with $\theta_1, \theta_2 \in \Theta$ and densities $p_1(\cdot), p_2(\cdot)$, we have

$$\inf_{\hat{\theta}} \sup_{\mathbb{P} \in \mathcal{F}} \mathbb{E}_{S \sim \mathbb{P}}[d(\hat{\theta}(S), \theta)] \geq \frac{d(\theta_1, \theta_2)}{4} \cdot \int_S \min(p_1(S), p_2(S))$$

The $\int_S$ term can be further lower bounded by $\frac{1}{2} \exp(-\mathrm{KL}(\mathbb{P}_1; \mathbb{P}_2))$ or other options [11]. We can relate the regret of some policy $\pi$ on a bandit instance $\theta$ to the estimation error of some estimator $\hat{\theta}(\cdot)$ on the trajectory distribution $\mathbb{P}_{\pi,\theta}$. The key to establishing this bound is to identify two instances $(\mathbb{P}_1, \theta_1)$ and $(\mathbb{P}_2, \theta_2)$, along with some common event, e.g. $D := \{d(\hat{\theta}(S), \theta_1) \geq d(\hat{\theta}(S), \theta_2)\}$, such that the sum of the two estimation errors can be lower bounded independent of $\hat{\theta}(\cdot)$:

$$\int_S p_1(S) \cdot d(\hat{\theta}(S), \theta_1) + \int_S p_2(S) \cdot d(\hat{\theta}(S), \theta_2) \geq \int_S \left[ p_1(S) \mathbf{1}\{D\} + p_2(S) \mathbf{1}\{\overline{D}\} \right] \cdot \frac{d(\theta_1, \theta_2)}{2}$$

**KL-divergence Decomposition.** Let $\mathbb{P}_{\theta_1}$ and $\mathbb{P}_{\theta_2}$ be the two trajectory distributions induced by the policy $\pi$ playing in the bandit instances $\theta_1$ and $\theta_2$, respectively. Consider a trajectory $A_1 y_2 \ldots A_n y_n$, the log-likelihood ratio $\log[p_{\theta_1}(A_1 \ldots y_n)/p_{\theta_2}(A_1 \ldots y_n)]$ equals $\sum_{t=1}^n \log[p_{\theta_1}(y_t|A_t)/p_{\theta_2}(y_t|A_t)]$. This allows us to decompose $\mathrm{KL}(\mathbb{P}_{\theta_1}; \mathbb{P}_{\theta_2})$ to $n$ KL-divergences between reward distributions conditioned on $A_t$ under $\mathbb{P}_{\theta_1}$. If reward distributions are $\mathcal{N}(\langle A_t, \theta_1 \rangle, 1)$ and $\mathcal{N}(\langle A_t, \theta_2 \rangle, 1)$, we have

$$\mathrm{KL}(\mathbb{P}_{\theta_1}; \mathbb{P}_{\theta_2}) = \mathbb{E}_{\theta_1} \left[ \sum_{t=1}^n \frac{1}{2} \langle A_t, \theta_1 - \theta_2 \rangle^2 \right]$$

# 3 Main results

In this section, we present our general succinctness model which encompasses cases such as entry-sparse vectors, group-sparse matrices and low-rank matrices. We then present a general minimax lower bound based on the proposed succinctness model.

## 3.1 Succinctness model

Here, we give the formal definition of vectors being "$s$-succinct" and related lemmas, in preparation for the lower bound construction in the next subsection.

We assume that all the 1-succinct vectors are given in advance and that the scalar multiple $aX$ of some succinct $X \in \mathbb{V}$ should be equally succinct at least when $a \neq 0$. We first impose the existence of the set of "succinct units", i.e. the 1-succinct vectors of unit length.

**Axiom 3.1** (existence of the succinct unit set). *There exists a non-empty set $\mathcal{U}$ such that $\forall E : E \in \mathcal{U} \Leftrightarrow E$ is an 1-succinct vector in $\mathbb{V}$ with $\|E\| = 1$, and that $\forall E : E \in \mathcal{U} \Rightarrow -E \in \mathcal{U}$.*

We aim to compose from some succinct units a new vector that can be regarded as "$s$-succinct" without any part or the whole reducing to "1-succinct". Then, those units the "$s$-succinct" vector decompose to must uphold some relation against all the succinct units in $\mathcal{U}$ to prove that they are indeed irreducible.

**Definition 3.1** (succinct support). We say that an indexed set of $d \in \mathbb{N}^+$ succinct units $\{E_i\}_{i=1}^d \subseteq \mathcal{U}$ forms a support if and only if

$$\sup_{E \in \mathcal{U}} \sum_{i=1}^d |\langle E, E_i \rangle| = 1 \tag{2}$$

*Remark.* It is easy to see that any non-empty subset of $\{E_i\}_{i=1}^d$ forms a support. We can flip the sign of any support member and still get a valid support, e.g. $\{\pm E_i\}_{i=1}^d$. The equality (2) implies mutual orthogonality. On the other hand, if $\mathcal{U}$ contains only mutually orthogonal elements, then $\mathcal{U}$ itself forms a support.

Now, we introduce the two quantities $Q(\cdot), R(\cdot) : \mathbb{V} \to [0, \infty)$ for later imposing boundedness condition, and then state the, by now, fairly evident definition of "$s$-succinct".

**Definition 3.2** (two semi-norms). For all $X \in \mathbb{V}$, we define $Q(X) := \sup_{E \in \mathcal{U}} \langle X, E \rangle$, as well as $R(X) := \sup_{Q(Y) \leq 1} \langle X, Y \rangle$ where $Y \in \mathbb{V}$ with $Q(Y) \leq 1$.

*Remark.* Since $\mathcal{U}$ is non-empty, both $Q(\cdot)$ and $R(\cdot)$ are validly defined and satisfy the triangle inequality. Moreover, because $\mathcal{U}$ is closed under additive inverse, it follows that $Q(X)$ and $R(X)$ also satisfy the property of absolute scalability. Hence, they qualify as semi-norms, as they satisfy all the norm properties except positive definiteness.

*Remark.* Given $X \neq \vec{0}$, $Q(X) = 0$ suggests $\forall E \in \mathcal{U} : \langle X, E \rangle = 0$. Thus, $Q(\cdot)$ will be a norm if $\mathcal{U}$ spans $\mathbb{V}$, as $\mathcal{U}$ will have no other orthogonal complement in $\mathbb{V}$ than $\vec{0}$. In addition, $R(X)$ will also be a norm because $R(X) \geq \langle X, X/Q(X) \rangle = \|X\|^2/Q(X) > 0$ as long as $X \neq 0$.

**Definition 3.3** ($s$-succinct). Consider $s \in \mathbb{N}^+$. We call a vector $X \in \mathbb{V}$ $s$-succinct if and only if it can can decompose to some support $\{E_i\}_{i=1}^s$ with some scalar coefficients $\{a_i\}_{i=1}^s$, i.e. $X = \sum_{i=1}^s a_i E_i$, and strictly $s$-succinct if and only if $a_i \neq 0$ for all $i \in [s]$.

Based on the definitions above, we have the following lemmas. See their proofs at Appendix A.

**Lemma 3.1.** *If $X = \sum_{i=1}^s a_i E_i$ for some support $\{E_i\}_{i=1}^s$ and some scalar coefficients $\{a_i\}_{i=1}^s$, then $Q(X) = \max_{i \in [s]} |a_i|$.*

**Lemma 3.2.** *If $X = \sum_{i=1}^s a_i E_i$ for some support $\{E_i\}_{i=1}^s$ and some scalar coefficients $\{a_i\}_{i=1}^s$, then $R(X) = \sum_{i=1}^s |a_i|$.*

**Lemma 3.3.** *If $X$ is simultaneously $s$-succinct and strictly $z$-succinct, then $s \geq z$.*

**Lemma 3.4.** *If $X$ is simultaneously strictly $s$-succinct and strictly $z$-succinct, then $s = z$.*

**Lemma 3.5.** *If $X$ is $s$-succinct, then $|\langle X, Y \rangle| \leq \min\left(Q(X)R(Y), Q(Y)R(X)\right)$ holds for any $Y \in \mathbb{V}$.*

**Lemma 3.6.** *If $X$ is $s$-succinct, then $\sup_{R(Y) \leq 1} \langle X, Y \rangle = Q(X)$, where $Y \in \mathbb{V}$ and $R(Y) \leq 1$.*

*Remark.* If $X$ is $s$-succinct, Lemma 3.1 and Lemma 3.2 ensure positive definiteness, i.e. $X \neq 0 \Rightarrow Q(X), R(X) > 0$. Meanwhile, Lemma 3.5 and Lemma 3.6 show that $Q(\cdot)$ and $R(\cdot)$ resemble a norm and its dual norm at least for $s$-succinct vectors. On the other hand, If $\mathcal{U}$ spans $\mathbb{V}$, then $Q(\cdot)$ becomes a norm and $R(\cdot)$ its dual norm. In this case, we have $|\langle X, Y \rangle| \leq Q(X)R(Y)$ and $\sup_{R(Y) \leq 1} \langle X, Y \rangle = Q(X)$ hold for all $X, Y \in \mathbb{V}$. However, $\mathcal{U}$ spanning $\mathbb{V}$ does not imply that every vector in $\mathbb{V}$ is $s$-succinct for some respective $s \in \mathbb{N}^+$.

## 3.2 General minimax lower bound

In what follows, we present our general lower bound for stochastic bandits under the succinctness model previously described. We start with a set of assumptions, which are fulfilled by our applications in Section 4.

**Assumption 3.7.** Suppose the following objects exist.

1. an action set $\mathcal{H} \subseteq \mathbb{V}$ satisfying $\max_{X \in \mathcal{H}} \langle X, \theta_0 \rangle \leq -C_0$, where $C_0 > 0$ is some global constant and $\theta_0 \in \mathbb{V}$ is some parameter that can decompose to some support $\{E_1', E_2', \dots, E_k'\}$ of cardinality $k \geq 1$.

2. $s - k$ non-empty groups of succinct units, denoted as $\mathcal{G}_1, \mathcal{G}_2, \dots, \mathcal{G}_{s-k}$, where for any tuple $(E_1, E_2, \dots, E_{s-k})$ in the Cartesian set $\bigtimes_{i=1}^{s-k} \mathcal{G}_i$, the set $\{E_1, \dots, E_{s-k}, E_1', \dots, E_k'\}$ forms a support of cardinality $s \geq 3k + 3$.

3. $q, p \in \mathbb{R}$ satisfying $\max_{X \in \mathcal{H}} \Phi_{\mathcal{G}_i}(X) \leq \frac{1}{q}$ and $\max_{X \in \mathcal{G}_i} \Phi_{\mathcal{G}_i}(X) \leq \frac{1}{p}$ for each $i \in [s-k]$, where $\Phi_{\mathcal{G}_i}(X) := \sum_{E \in \mathcal{G}_i} \langle X, E \rangle^2 / |\mathcal{G}_i|$.

Next, we introduce our main theoretical result. We provide a formal proof based on a sequence of steps and some claims that are proved in the Appendix B.

**Theorem 3.8.** *Consider the bandit problem* (1) *where all actions $X \in \mathbb{V}$ and parameters $\theta \in \mathbb{V}$ are bounded with $Q(X) \leq 1$ and $R(\theta) \leq 1$. Suppose Assumption 3.7 hold with some $s \in \mathbb{N}^+$, $C_0 > 0$*

*and $p, q \geq 1$. Then, we can construct an action set $\mathcal{A}$ such that given any policy $\pi$, there exists an $s$-succinct parameter $\theta$ to incur regret*

$$R_n^\pi(\mathcal{A}, \theta) \geq \frac{\min(C_0, e^{-4}/8)}{3} \cdot \min(s^{\frac{2}{3}} n^{\frac{2}{3}} q^{\frac{1}{3}}, s\sqrt{pn}) \tag{3}$$

*Proof.* Since Assumption 3.7 holds, we assume access to $\theta_0$, the action set $\mathcal{H}$, and the $s - k$ groups $\mathcal{G}_1, \mathcal{G}_2, \ldots, \mathcal{G}_{s-k}$. Let $b = \lfloor (s - k)/2 \rfloor$. It follows that $b \geq k + 1$ and $3b \geq 2b + k + 1 \geq s > 2b$. Note that $0 < C_0 \leq 1$ due to $R(\theta_0) \leq 1$ and Lemma 3.5. We choose a specific $(E_1^*, E_2^*, \ldots, E_b^*) \in \times_{i=1}^b \mathcal{G}_{b+i}$ and extend each of the first $b$ groups with a corresponding star-marked unit as follows:

$$\mathcal{G}_1 \cup \{E_1^*\}, \quad \mathcal{G}_2 \cup \{E_2^*\}, \quad \ldots, \quad \mathcal{G}_b \cup \{E_b^*\}$$

**Step 1: Construction of action set $\mathcal{A}$.**   We construct

$$\mathcal{S} := \left\{ \sum_{i=1}^b \tilde{E}_i \,\middle|\, \forall i \in [b]: \; \tilde{E}_i \in \mathcal{G}_i \cup \{E_i^*\} \right\}, \qquad \mathcal{A} := \mathcal{H} \cup \mathcal{S}$$

Each action $X \in \mathcal{S}$ is $s$-succinct and satisfies $Q(X) = 1$ by Lemma 3.1. Since each action corresponds with a unique selection of a unit $\tilde{E}_i$ from each extended-group $\mathcal{G}_i \cup \{E_i^*\}$, later when discussing the action $A_t$ at round $t$ and if $A_t \in \mathcal{S}$, we will use the notation $\tilde{A}_{t,i} := \tilde{E}_i$ to represent its unit selection in the extended-group $i$.

**Step 2: Parameters in consideration.**   Let $\varepsilon > 0$ be a constant to be decided later, subject to the constraint $s\varepsilon \leq \frac{C_0}{3}$. We construct

$$\Theta := \left\{ \frac{1}{2}\theta_0 + \varepsilon \sum_{i=1}^b \left( E_i^* + 2\tilde{E}_i \cdot \mathbf{1}\{\tilde{E}_i \neq E_i^*\} \right) \,\middle|\, \forall i \in [b]: \; \tilde{E}_i \in \mathcal{G}_i \cup \{E_i^*\} \right\}$$

We can verify that $\forall \theta \in \Theta : R(\theta) \leq \frac{1}{2} + 3b\varepsilon \leq 1$. Similar to the construction of $\mathcal{S}$, there is a one-to-one correspondence between $\Theta$ and $\times_{i=1}^b (\mathcal{G}_i \cup \{E_i^*\})$. For any $\theta \in \Theta$ and $i \in [b]$, we use $\tilde{E}_i(\theta) \in \mathcal{G}_i \cup \{E_i^*\}$ to indicate the unit that $\theta$ selects in the extended group $i$.

By construction, playing any $\mathcal{H}$-based action in any parameter setting $\theta$ incurs at least a constant regret per round, compared to an optimal $\mathcal{S}$-based action that shares the same unit selection as $\theta$. By constraining $s\varepsilon$ to be sufficiently small relative to $C_0$, we can show that $\forall \theta \in \Theta, \forall X \in \mathcal{H}$: $\Delta_\theta(X) \geq C_0/3$. See the formal argument at Appendix B.1.

**Step 3: Intricacies of avoiding sub-optimality.**   Choosing the correct optimal $\mathcal{S}$-based action across different parameter settings of $\theta$ can be seen as solving $b$ sub-problems in parallel, where the challenge is to minimize the regret of failing to select the correct unit $\tilde{E}_i(\theta)$ for each extended-group $i \in [b]$. Define $T_\mathcal{S} := \{t \in [n] \,|\, A_t \in \mathcal{S}\}$, For each $i \in [b]$, we introduce an event $D_i$ (and its complement event $\overline{D}_i$):

$$D_i = \left\{ \sum_{t \in T_\mathcal{S}} \mathbf{1}\{\tilde{A}_{t,i} = E_i^*\} \geq \frac{n}{2} \right\}$$

**Claim 3.9.** $\forall \theta \in \Theta$, we can lower bound the regret in terms of those group-wise events:

$$R_n^\pi(\mathcal{A}, \theta) \geq \sum_{i=1}^b R_i^\theta, \quad \text{where } R_i^\theta := \frac{n\varepsilon}{2} \cdot \begin{cases} \mathbb{P}_\theta(\overline{D}_i) & \text{if } \tilde{E}_i(\theta) = E_i^* \\ \mathbb{P}_\theta(D_i) & \text{if } \tilde{E}_i(\theta) \neq E_i^* \end{cases}$$

where $\mathbb{P}_\theta$ is the distribution over empirical trajectories induced by $\pi$ playing in $\theta$ up to $n$ rounds. See the proof at Appendix B.1.

**Step 4: Bound through KL-divergence.** Intuitively, the sparse geometry in $\mathcal{S}$ and $\Theta$ within each extended-group $i$ puts an information-theoretic limit on $\pi$'s ability to distinguish between the situations $\tilde{E}_i(\theta) = E^*$ and $\tilde{E}_i(\theta) \neq E^*$ over all versions of $\theta$.

We consider different settings of $\theta$ with varying $\tilde{E}_i(\theta)$ for the extended-group $i$, while keeping their unit selection fixed for the other extended-groups. Given $(\tilde{E}_1, \tilde{E}_2, \ldots, \tilde{E}_b) \in \bigtimes_{i=1}^{b} (\mathcal{G}_i \cup \{E_i^*\})$ and extended-group index $i$, we reversely denote $\theta(\tilde{E}_i, \tilde{\mathbf{E}}_{-i}) := \frac{1}{2}\theta_0 + \varepsilon \sum_{i=1}^{b}(E_i^* + 2\tilde{E}_i \cdot \mathbf{1}\{\tilde{E}_i \neq E_i^*\})$, where $\tilde{\mathbf{E}}_{-i} := (\tilde{E}_1, \ldots, \tilde{E}_{i-1}, \tilde{E}_{i+1}, \ldots, \tilde{E}_b)$. Fix $i$ and $\tilde{\mathbf{E}}_{-i}$. Then, using Le Cam's Lemma, KL-divergence decomposition, and Jensen's Inequality, we can lower bound the following quantity:

$$\sum_{E \in \mathcal{G}_i} R_i^{\theta(E_i^*, \tilde{\mathbf{E}}_{-i})} + R_i^{\theta(E, \tilde{\mathbf{E}}_{-i})} \geq \frac{n\varepsilon|\mathcal{G}_i|}{4} \exp\left(-2\varepsilon^2 \mathbb{E}_{\theta(E_i^*, \tilde{\mathbf{E}}_{-i})}\left[\sum_{t=1}^{n}\sum_{E \in \mathcal{G}_i}\frac{\langle A_t, E\rangle^2}{|\mathcal{G}_i|}\right]\right)$$

This intermediate bound is established to support the following claim. See the details at Appendix B.2.

**Claim 3.10.** Define $T_{\mathcal{H}} := \{t \in [n] \mid A_t \in \mathcal{H}\}$. We have

$$\max_{\theta \in \Theta} R_n^\pi(\mathcal{A}, \theta) \geq \frac{ns\varepsilon}{24} \exp\left(-2\varepsilon^2 \left[\frac{n}{p} + \frac{1}{q} \cdot \max_{\theta \in \Theta} \mathbb{E}_\theta[|T_{\mathcal{H}}|]\right]\right) \tag{4}$$

**Step 5: Information-regret tradeoff.** Overall, the lower bound (4) is optimistic regarding $\pi$ playing more $\mathcal{H}$-based actions, as $\mathcal{S}$-based actions are sparse and provide limited information for distinguishing between different $\theta$ across $\Theta$. However, since $\mathcal{H}$-based actions are also sub-optimal, there is another lower bound on the maximal regret that discourages the same quantity $\max_{\theta \in \Theta} \mathbb{E}_\theta[|T_{\mathcal{H}}|]$:

$$\max_{\theta \in \Theta} R_n^\pi(\mathcal{A}, \theta) \geq \max_{\theta \in \Theta} \mathbb{E}_\theta\left[\sum_{t \in T_{\mathcal{H}}} \Delta_\theta(A_t)\right] \geq \frac{C_0}{3} \cdot \max_{\theta \in \Theta} \mathbb{E}_\theta[|T_{\mathcal{H}}|] \tag{5}$$

The challenge of determining how often to play regrettable yet potentially informative $\mathcal{H}$-based actions leads us to the final lower bound. Combining the lower bounds (4) and (5) in a minimum expression, we can replace $\max_{\theta \in \Theta} \mathbb{E}_\theta[|T_{\mathcal{H}}|]$ with a free variable $h$, and obtain the lower bound $\min\left(\frac{ns\varepsilon}{24}\exp(-2\varepsilon^2[\frac{n}{p} + \frac{h}{q}]), \frac{C_0}{3}h\right)$ that holds for any $h \in \mathbb{R}$.

When $p < n^{\frac{1}{3}}s^{-\frac{2}{3}}q^{\frac{2}{3}}$, we let $\varepsilon = \sqrt{p/n}$ and $h = q\varepsilon^{-2}$ (the constraint $s\varepsilon \leq C_0/3$ requires $9s^2p \leq C_0^2 n$). Then, we have a lower bound $\min(e^{-4}/24, C_0/3) \cdot s\sqrt{pn}$. When $p \geq n^{\frac{1}{3}}s^{-\frac{2}{3}}q^{\frac{2}{3}}$, we let $\varepsilon = n^{-\frac{1}{3}}s^{-\frac{1}{3}}q^{\frac{1}{3}}$ and $h = q\varepsilon^{-2}$ ( $s\varepsilon \leq C_0/3$ requires $27s^2q \leq C_0^3 n$). Then, we have another lower bound $\min(e^{-4}/24, C_0/3) \cdot s^{\frac{2}{3}}n^{\frac{2}{3}}q^{\frac{1}{3}}$. Together, we have the final lower bound (3). $\qquad\square$

## 4 Applications

In this section, we show that our framework sharpens a recent lower bound by [7] when applied to entry-sparse vectors. We then apply our framework to provide the first lower bounds (to the best of our knowledge) for the group-sparse and low-rank matrix settings.

First, we derive corollaries for bandit problems with three types of succinct representations: entry-sparse vector, group-sparse matrix, and low-rank matrix. Accordingly, we consider $\mathbb{V} := \mathbb{R}^d$ or $\mathbb{R}^{d_1 \times d_2}$, treating vectors in $\mathbb{R}^d$ as one-column matrices.

Naturally, the inner product is defined as $\langle X, Y \rangle := \text{trace}(X^T Y)$ for all matrices $X, Y \in \mathbb{V}$ and the induced norm $\|X\| := \sqrt{\langle X, X \rangle}$ follows. In addition, we define the following quantity to characterize the shape of an action set $\mathcal{A} \subseteq \mathbb{V}$.

**Definition 4.1.** Let $\text{Pr}(\mathcal{A})$ be the space of probability distributions over $\mathcal{A}$ and define

$$C_{\min}(\mathcal{A}) := \max_{\mu \in \text{Pr}(\mathcal{A})} \min_{\beta} \mathbb{E}_{X \sim \mu}[\langle X, \beta\rangle^2] \quad \text{where } \beta \in \mathbb{V} \text{ with } \|\beta\| = 1$$

*Remark.* By flattening matrices $X \sim \mu$ into vectors $x$, $C_{\min}(\mathcal{A})$ can be computed by finding a distribution $\mu$ that maximizes the minimum eigenvalue of the covariance matrix $\mathbb{E}_{x \sim \mu}[xx^T]$. Intuitively speaking, $C_{\min}(\cdot)$ measures how well the set $\mathcal{A}$ can support an exploration distribution that samples well-conditioned action data for probing the unknown bandit parameter.

*Remark.* $C_{\min}(\mathcal{A}) > 0$ if and only if $\mathcal{A}$ spans $\mathbb{V}$. $C_{\min}(\mathcal{A})$ is upper bounded by 1 if actions in $\mathcal{A}$ are bounded with $\|\cdot\|_\infty \leq 1$, by $1/d_2$ if $\|\cdot\|_{\infty,2} \leq 1$, and by $1/\min(d_1, d_2)$ if $\|\cdot\|_{op} \leq 1$.

## 4.1 Entry-sparse vector bandits

First, we use our general framework to obtain a lower bound of order $\Omega(\min(C_{\min}^{-1/3} s^{2/3} n^{2/3}, \sqrt{dsn}))$ for entry-sparse vectors. Our result is tighter than the prior rate of $\Omega(\min(C_{\min}^{-1/3} s^{1/3} n^{2/3}, \sqrt{dn}))$ shown in [7].

**Corollary 4.1.** *Consider the bandit problem* (1) *where actions* $x \in \mathbb{R}^d$ *and parameters* $\theta \in \mathbb{R}^d$ *are bounded with* $\|x\|_\infty \leq 1$ *and* $\|\theta\|_1 \leq 1$. *Given any policy* $\pi$, *there exist an action set* $\mathcal{A}$ *with* $C_{\min}(\mathcal{A})$ *and a parameter* $\theta$ *with no more than* $s$ *non-zero entries, such that*

$$R_n^\pi(\mathcal{A}, \theta) \geq \frac{\exp(-4)}{24} \cdot \min\left(C_{\min}^{-\frac{1}{3}}(\mathcal{A}) s^{\frac{2}{3}} n^{\frac{2}{3}}, \sqrt{dsn}\right) \tag{6}$$

*Proof.* We define the succinct unit set $\mathcal{U} := \{e_j\}_{j=1}^d$, where each $e_j := (0, 0, \ldots, 1, \ldots, 0)^T \in \mathbb{R}^d$ has a single 1 at the $j$-th entry. Then, the two semi-norms $Q(\cdot) = \|\cdot\|_\infty$ and $R(\cdot) = \|\cdot\|_1$. Meanwhile, $\theta$ having no more than $s$ non-zero entries is equivalent to it being $s$-succinct. Let $0 < \kappa \leq 1$ be a constant and construct the action set

$$\mathcal{H} := \left\{ \sum_{j=1}^d a_j \cdot e_j \;\middle|\; \forall j \in [d] : a_j = \begin{cases} 1 & \text{if } j = 1 \\ \pm\kappa & \text{otherwise} \end{cases} \right\}$$

We consider $\theta_0 := -e_1$ which is 1-succinct and satisfies $\max_{x \in \mathcal{H}} \langle x, \theta_0 \rangle \leq -1$. It also can be shown that $C_{\min}(\mathcal{H}) \geq \kappa^2$. Assume $d \pmod{s} = 0$ and construct

$$\forall i \in [s-1] : \; \mathcal{G}_i := \left\{ e_j \;\middle|\; j \in \left[\frac{id}{s} + 1 : \frac{id}{s} + \frac{d}{s}\right] \right\}$$

We can verify that $\max_{x \in \mathcal{G}_i} \Phi_{\mathcal{G}_i}(x) \leq s/d$ and $\max_{x \in \mathcal{H}} \Phi_{\mathcal{G}_i}(x) \leq \kappa^2$ for each $i \in [s-1]$. Invoking Theorem 3.8 with the constructed $\mathcal{H}$ and $\{\mathcal{G}_i\}_{i=1}^{s-1}$, we can construct an action set $\mathcal{A}$ with $C_{\min}(\mathcal{A}) \geq C_{\min}(\mathcal{H})$ and $s$-succinct $\theta$ that make the lower bound (6) hold. $\square$

## 4.2 Group-sparse matrix bandits

Next, we use our general framework to obtain the first lower bound for group-sparse matrices. The following corollary provides a lower bound of order $\Omega(\min(C_{\min}^{-1/3} s^{2/3} n^{2/3}, \sqrt{d_1 d_2 sn}))$.

**Corollary 4.2.** *Consider the bandit problem* (1) *where actions* $X \in \mathbb{R}^{d_1 \times d_2}$ *and parameters* $\theta \in \mathbb{R}^{d_1 \times d_2}$ *are bounded with* $\|X\|_{\infty,2} \leq 1$ *and* $\|\theta\|_{1,2} \leq 1$. *Given any policy* $\pi$, *there exist an action set* $\mathcal{A}$ *with constant* $C_{\min}(\mathcal{A})$ *and a parameter* $\theta$ *with no more than* $s$ *non-zero rows, such that*

$$R_n^\pi(\mathcal{A}, \theta) \geq \frac{\exp(-4)}{24} \cdot \min\left(C_{\min}^{-\frac{1}{3}}(\mathcal{A}) s^{\frac{2}{3}} n^{\frac{2}{3}}, \sqrt{d_1 d_2 sn}\right) \tag{7}$$

*Proof.* We define the succinct unit set $\mathcal{U} := \{ev^T \,|\, e \in \{e_j\}_{j=1}^{d_1}, v \in \mathbb{R}^{d_2}, \|v\| = 1\}$, where $\{e_j\}_{j=1}^{d_1}$ is the standard basis of $\mathbb{R}^{d_1}$. Then, the two semi-norms $Q(\cdot) = \|\cdot\|_{\infty,2}$ and $R(\cdot) = \|\cdot\|_{1,2}$. Meanwhile, $\theta$ having no more than $s$ non-zero rows is equivalent to it being $s$-succinct. Let $\{v_g\}_{g=1}^{d_2}$ be an orthonormal basis of $\mathbb{R}^{d_2}$ and $0 < \kappa \leq \frac{1}{2}$ be a constant. We construct the action set

$$\mathcal{H} := \bigcup_{g \in [d_2]} \left\{ \frac{1}{2} e_1 v_1^T + \sum_{j=1}^{d_1} a_j \cdot e_j v_g^T \;\middle|\; \forall j \in [d_1] : a_j = \begin{cases} 0 & \text{if } j = g = 1 \\ \pm\kappa & \text{otherwise} \end{cases} \right\}$$

We consider $\theta_0 := -e_1 v_1^T$ which is 1-succinct and satisfies $\max_{X \in \mathcal{H}} \langle X, \theta_0 \rangle \leq -\frac{1}{2}$. It also can be shown that $C_{\min}(\mathcal{H}) \geq \kappa^2/d_2$. Assume $d_1 \pmod{s} = 0$ and construct

$$\forall i \in [s-1]: \ \mathcal{G}_i := \left\{ e_j v_m^T \ \middle| \ j \in \left[ \frac{id_1}{s} + 1 : \frac{id_1}{s} + \frac{d_1}{s} \right], m \in [d_2] \right\}$$

We can verify that $\max_{x \in \mathcal{G}_i} \Phi_{\mathcal{G}_i}(x) \leq s/(d_1 d_2)$ and $\max_{x \in \mathcal{H}} \Phi_{\mathcal{G}_i}(x) \leq \kappa^2/d_2$ for each $i \in [s-1]$. Invoking Theorem 3.8 with the constructed $\mathcal{H}$ and $\{\mathcal{G}_i\}_{i=1}^{s-1}$, we can construct an action set $\mathcal{A}$ with $C_{\min}(\mathcal{A}) \geq C_{\min}(\mathcal{H})$ and $s$-succinct $\theta$ that make the lower bound (7) hold. $\qquad \square$

### 4.3 Low-rank matrix bandits

In what follows, we use our general framework to obtain the first lower bound for a slightly more challenging problem: low-rank matrices. The corollary below provides a lower bound of order $\Omega(\min(C_{\min}^{-1/3} s^{1/3} n^{2/3}, \sqrt{d_1 d_2 n}))$.

**Corollary 4.3.** *Consider the bandit problem* (1) *where actions $X \in \mathbb{R}^{d_1 \times d_2}$ and parameters $\theta \in \mathbb{R}^d$ are bounded with $\|X\|_{\mathrm{op}} \leq 1$ and $\|\theta\|_{\mathrm{nuc}} \leq 1$. Given any policy $\pi$, there exist an action set $\mathcal{A}$ with constant $C_{\min}(\mathcal{A})$ and a parameter $\theta$ with its rank no larger than $s$, such that*

$$R_n^\pi(\mathcal{A}, \theta) \geq \frac{\exp(-4)}{24} \cdot \min\left( C_{\min}^{-\frac{1}{3}}(\mathcal{A}) s^{\frac{1}{3}} n^{\frac{2}{3}}, \sqrt{d_1 d_2 n} \right) \tag{8}$$

*Proof.* We define the succinct unit set $\mathcal{U} := \{uv^T \mid u \in \mathbb{R}^{d_1}, v \in \mathbb{R}^{d_2}, \|u\| = \|v\| = 1\}$. Then, the two semi-norms $Q(\cdot) = \|\cdot\|_{\mathrm{op}}$ and $R(\cdot) = \|\cdot\|_{\mathrm{nuc}}$. Meanwhile, $\theta$ having a rank no larger than $s$ is equivalent to it being $s$-succinct. Let $\{u_j\}_{j=1}^{d_1}$ and $\{v_g\}_{g=1}^{d_2}$ be orthonormal bases of $\mathbb{R}^{d_1}$ and $\mathbb{R}^{d_2}$, respectively. Let $0 < \kappa \leq \frac{1}{2}$ be a constant and assume $d_1 \leq d_2$ (the case $d_1 > d_2$ can be handled by swapping the subscripts of $u$ and $v$ below). We construct the action set

$$\mathcal{H} := \bigcup_{g \in [d_2]} \left\{ \frac{1}{2} u_1 v_1^T + \sum_{j=1}^{d_1} a_j \cdot u_j v_{m(g,j)}^T \ \middle| \ \forall j \in [d_1]: a_j = \begin{cases} 0 & \text{if } j = m(g,j) = 1 \\ \pm\kappa & \text{otherwise} \end{cases} \right\}$$

where $m(g,j) := (g+j) \pmod{d_2} + 1$ is used to cyclically iterate over $[d_2]$ for each $j \in [d_1]$, with a different starting point for each $g \in [d_2]$. We consider $\theta_0 := -e_1 v_1^T$ which is 1-succinct and satisfies $\max_{X \in \mathcal{H}} \langle X, \theta_0 \rangle \leq -\frac{1}{2}$. It also can be shown that $C_{\min}(\mathcal{H}) \geq \kappa^2/d_2$. Assume $d_1 \pmod{s} = d_2 \pmod{s} = 0$ and construct

$$\forall i \in [s-1]: \ \mathcal{G}_i := \left\{ u_j v_m^T \ \middle| \ j \in \left[ \frac{id_1}{s} + 1 : \frac{id_1}{s} + \frac{d_1}{s} \right], m \in \left[ \frac{id_2}{s} + 1 : \frac{id_2}{s} + \frac{id_2}{s} \right] \right\}$$

We can verify that $\max_{x \in \mathcal{G}_i} \Phi_{\mathcal{G}_i}(x) \leq s^2/(d_1 d_2)$ and $\max_{x \in \mathcal{H}} \Phi_{\mathcal{G}_i}(x) \leq \kappa^2 s/d_2$ for each $i \in [s-1]$. Invoking Theorem 3.8 with the constructed $\mathcal{H}$ and $\{\mathcal{G}_i\}_{i=1}^{s-1}$, we can construct an action set $\mathcal{A}$ with $C_{\min}(\mathcal{A}) \geq C_{\min}(\mathcal{H})$ and $s$-succinct $\theta$ that make the lower bound (8) hold. $\qquad \square$

## 5 Future Work

In this paper we give a first step towards unifying the study of lower bounds for succinct stochastic bandits. Although we provide three applications including two novel lower bounds, we believe our framework will motivate future work for finding lower bounds on other bandit problems. The study of other stochastic bandits over mathematical objects beyond vectors and matrices, such as a multi-linear regimes (e.g., tensors) and kernels of combinatorial objects (e.g., sentences, trees) would be interesting. In addition, the study of nonparametric settings, in which the bases are functions would also be an interesting extension.

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

# Supplementary Material: A Novel General Framework for Sharp Lower Bounds in Succinct Stochastic Bandits

## A  Proofs of succinctness-related lemmas

Here, we provide detailed proofs for the succinctness-related lemmas in the main text.

### A.1  Proof of Lemma 3.1

We rewrite $X = \sum_{i=1}^{s} a_i E_i$ as $X = \sum_{i=1}^{s} |a_i| \cdot \text{sign}(a_i) E_i$[1]. Then,

$$
\begin{aligned}
\sup_{E \in \mathcal{U}} \langle X, E \rangle &= \sup_{E \in \mathcal{U}} \sum_{i=1}^{s} |a_i| \cdot \langle \text{sign}(a_i) E_i, E \rangle \\
&\leq \sup_{E \in \mathcal{U}} \max_{i \in [s]} |a_i| \cdot \sum_{i=1}^{s} |\langle \text{sign}(a_i) E_i, E \rangle| \\
&= \max_{i \in [s]} |a_i| \cdot \sup_{E \in \mathcal{U}} \sum_{i=1}^{s} |\langle E_i, E \rangle| \\
&= \max_{i \in [s]} |a_i| \tag{9}
\end{aligned}
$$

Notably, step (9) applies the succinct support requirement (2) on $\{E_i\}_{i=1}^{s}$. This upper bound is reachable by letting $E = \text{sign}(a_m) E_m$ where $m \in \arg\max_{i \in [s]} |a_i|$. Since $\text{sign}(a_m) E_m \in \mathcal{U}$, we have $Q(X) := \sup_{E \in \mathcal{U}} \langle X, E \rangle = \max_{i \in [s]} |a_i|$.

### A.2  Proof of Lemma 3.2

We rewrite $X = \sum_{i=1}^{s} a_i E_i$ as $X = \sum_{i=1}^{s} |a_i| \cdot \text{sign}(a_i) E_i$. Then,

$$
\begin{aligned}
\sup_{Q(Y) \leq 1} \langle X, Y \rangle &= \sup_{Q(Y) \leq 1} \sum_{i=1}^{s} |a_i| \cdot \langle \text{sign}(a_i) E_i, Y \rangle \\
&\leq \sup_{Q(Y) \leq 1} \sum_{i=1}^{s} |a_i| \cdot |\langle E_i, Y \rangle| \\
&\leq \sup_{Q(Y) \leq 1} \sum_{i=1}^{s} |a_i| \cdot \sup_{E \in \mathcal{U}} |\langle E, Y \rangle| \\
&= \sup_{Q(Y) \leq 1} \sum_{i=1}^{s} |a_i| \cdot Q(Y) \tag{10} \\
&\leq \sum_{i=1}^{s} |a_i| \tag{11}
\end{aligned}
$$

Equality (10) requires the fact that $\mathcal{U}$ is closed under flipping the sign of any $E \in \mathcal{U}$. The upper bound in (11) is reachable by letting $Y = \sum_{i=1}^{s} \text{sign}(a_i) E_i$, in which case $Q(Y) = 1$ due to Lemma 3.1. Therefore, we have $R(X) := \sup_{Q(Y) \leq 1} \langle X, Y \rangle = \sum_{i=1}^{s} |a_i|$.

---

[1] $\text{sign}(x) = \mathbf{1}\{x \geq 0\} - \mathbf{1}\{x < 0\}$

## A.3 Proof of Lemma 3.3

From the definition of $s$-succinct and $z$-succinct, we know that $X = \sum_{i=1}^{s} a_i E_i = \sum_{i=1}^{z} b_i E'_i$ for some $\{E_i\}_{i=1}^{s}$ and some $\{E'_i\}_{i=1}^{z}$ that constitute a succinct support respectively, and where the coefficients can fulfill $b_i \neq 0$ for all $i \in [z]$.

Let $Y = \sum_{i=1}^{s} \text{sign}(a_i) E_i$. Due to the mutual orthogonality of support members, we have $\langle X, Y \rangle = \langle \sum_{i=1}^{s} a_i E_i, Y \rangle = \sum_{i=1}^{s} |a_i|$ and $\langle Y, Y \rangle = s$. Importantly, $Q(Y) = 1$ according to Lemma 3.1.

Meanwhile, Lemma 3.2 suggests $R(X) = \sum_{i=1}^{s} |a_i| = \sum_{i=1}^{z} |b_i|$. Once again, from $X$'s decomposition as $\sum_{i=1}^{z} |b_i| \cdot \text{sign}(b_i) E'_i$, we calculate

$$\langle X, Y \rangle = \sum_{i=1}^{z} |b_i| \cdot \langle \text{sign}(b_i) E'_i, Y \rangle = \sum_{i=1}^{z} |b_i| \tag{12}$$

For each $i \in [z]$, we have an upper bound

$$\langle \text{sign}(b_i) E'_i, Y \rangle \leq |\langle E'_i, Y \rangle| \leq \sup_{E \in \mathcal{U}} |\langle E, Y \rangle| = \sup_{E \in \mathcal{U}} \langle E, Y \rangle = Q(Y) = 1 \tag{13}$$

Since $|b_i| > 0$ for all $i \in [z]$, for equality (12) to hold, the inequalities in (13) must hold with equality, meaning $|\langle E'_i, Y \rangle| = 1$, for all $i \in [z]$.

Let $P = \sum_{i=1}^{z} \langle E'_i, Y \rangle \cdot E'_i$ be the projection of $Y$ onto the set of bases $\{E'_i\}_{i=1}^{z}$. Then, by construction, $\langle P, Y \rangle = \langle P, P \rangle = \sum_{i=1}^{z} \langle E'_i, Y \rangle^2 = z$. Combining them with the Cauchy-Schwarz inequality $\|P\| \cdot \|Y\| \geq \langle P, Y \rangle$, we can conclude $\|Y\| = \sqrt{\langle Y, Y \rangle} = \sqrt{s} \geq z/\sqrt{z} = \sqrt{z}$, and thus $s \geq z$.

## A.4 Proof of Lemma 3.4

By definition, $X$ is also $s$-succinct and $z$-succinct at the same time. From Lemma 3.3, we have $s \geq z$ and $z \geq s$ at the same time. Naturally, $s = z$.

## A.5 Proof of Lemma 3.5

Recall that $Q(\cdot)$ is a semi-norm, meaning $Q(kA) = |k|Q(A)$ and $Q(A) \geq 0$ for all $k \in \mathbb{R}$ and $A \in \mathbb{V}$. Therefore, for any $A, B \in \mathbb{V}$ with $Q(A) \neq 0$

$$\frac{|\langle A, B \rangle|}{Q(A)} \leq \sup_{Q(A') \neq 0} \frac{|\langle A', B \rangle|}{Q(A')} \leq \sup_{Q(A') \leq 1} |\langle A', B \rangle| = \sup_{Q(A') \leq 1} \langle A', B \rangle = R(B)$$

By substituting $A, B$ with $X, Y$, we have $|\langle X, Y \rangle| \leq Q(X)R(Y)$ as long as $Q(X) \neq 0$. In the case of $Q(X) = 0$, given that $X = \sum_{i=1}^{s} a_i E_i$ for some support $\{E_i\}_{i=1}^{s}$ and some coefficients $\{a_i\}_{i=1}^{s}$, we have $Q(X) = \max_{i \in [s]} |a_i| = 0$ from Lemma 3.1, which leads to $X = \vec{0}$ and thus $\langle X, Y \rangle = 0 = Q(X)R(Y)$. Together, $|\langle X, Y \rangle| \leq Q(X)R(Y)$.

By substituting $A, B$ with $Y, X$, we have $|\langle X, Y \rangle| \leq Q(Y)R(X)$ as long as $Q(Y) \neq 0$. In the case of $Q(Y) := \sup_{E \in \mathcal{U}} \langle Y, E \rangle = 0$, it is easy to see that $\langle E_i, Y \rangle = 0$ for all the members in the support $\{E_i\}_{i=1}^{s}$ and thus $\langle X, Y \rangle = 0$. Together, $|\langle X, Y \rangle| \leq Q(Y)R(X)$.

## A.6 Proof of Lemma 3.6

Applying Lemma 3.5 within the maximization, we obtain the upper bound

$$\sup_{R(Y) \leq 1} \langle X, Y \rangle \leq \sup_{R(Y) \leq 1} R(Y)Q(X) \leq Q(X)$$

Given that $X = \sum_{i=1}^{s} a_i E_i$ for some support $\{E_i\}_{i=1}^{s}$ and some coefficients $\{a_i\}_{i=1}^{s}$, let $Y = \text{sign}(a_m) E_m$ where $m \in \arg\max_{i \in [s]} |a_i|$. We can verify that $\langle X, Y \rangle = \max_{i \in [s]} |a_i| = Q(X)$ due to Lemma 3.1, and that $R(Y) = 1$ due to Lemma 3.2. Therefore, $\sup_{R(Y) \leq 1} \langle X, Y \rangle = Q(X)$.

# B Proof details for Theorem 3.8

Here, we provide detailed proofs for the claims inside the proof of Theorem 3.8 in the main text.

## B.1 Proof of Claim 3.9

With $b\varepsilon \leq \frac{s\varepsilon}{2} \leq \frac{C_0}{6}$, we can prove that $\forall \theta \in \Theta, \forall X \in \mathcal{H}$:

$$
\begin{aligned}
\Delta_\theta(X) &\geq \max_{X \in \mathcal{A}} \langle X, \theta \rangle - \max_{X \in \mathcal{H}} \langle X, \theta \rangle \\
&\geq \max_{X \in \mathcal{S}} \langle X, \theta \rangle - \left[ \max_{X \in \mathcal{H}} \langle X, \frac{1}{2}\theta_0 \rangle + \max_{Q(X) \leq 1} \langle X, \theta - \frac{1}{2}\theta_0 \rangle \right] \\
&\geq \varepsilon \sum_{i=1}^{b} (1 + \mathbf{1}\{\tilde{E}_i(\theta) \neq E_i^*\}) - \left[ -\frac{C_0}{2} + \varepsilon \sum_{i=1}^{b} (1 + 2 \cdot \mathbf{1}\{\tilde{E}_i(\theta) \neq E_i^*\}) \right] \quad (14) \\
&= \frac{C_0}{2} - \varepsilon \sum_{i=1}^{b} \mathbf{1}\{\tilde{E}_i(\theta) \neq E_i^*\} \\
&\geq \frac{C_0}{3} \\
&\geq \varepsilon \sum_{i=1}^{b} \mathbf{1}\{\tilde{E}_i(\theta) = E_i^*\} \quad (15)
\end{aligned}
$$

Recall that by construction, $\theta - \frac{1}{2}\theta_0 = \varepsilon \sum_{i=1}^{b} E_i^* + 2\tilde{E}_i(\theta) \cdot \mathbf{1}\{\tilde{E}_i(\theta) \neq E_i^*\}$. In step (14), $\max_{X \in \mathcal{S}} \langle X, \theta \rangle$ is obtained by the action $X = \sum_{i=1}^{b} \tilde{E}_i(\theta)$, while $\max_{Q(X) \leq 1} \langle X, \theta - \frac{1}{2}\theta_0 \rangle$ can be achieved by an action $X = \sum_{i=1}^{b} E_i^* + \tilde{E}_i(\theta) \cdot \mathbf{1}\{\tilde{E}_i(\theta) \neq E_i^*\}$. Further, we have that $\forall \theta \in \Theta$ :

$$
\begin{aligned}
R_n^\pi(\theta) &:= \mathbb{E}_\theta \left[ \sum_{t=1}^{n} \max_{X \in \mathcal{A}} \langle X, \theta \rangle - \langle A_t, \theta \rangle \right] \\
&\geq \mathbb{E}_\theta \left[ \sum_{t=1}^{n} \begin{cases} \max_{X \in \mathcal{S}} \langle X, \theta \rangle - \langle A_t, \theta \rangle & \text{if } A_t \in \mathcal{S} \\ \varepsilon \sum_{i=1}^{b} \mathbf{1}\{\tilde{E}_i(\theta) = E_i^*\} & \text{if } A_t \in \mathcal{H} \end{cases} \right] \quad (16) \\
&= \sum_{t=1}^{n} \mathbb{E}_\theta \left[ \varepsilon \sum_{i=1}^{b} \begin{cases} 1 - \mathbf{1}\{\tilde{A}_{t,i} = E_i^*\} & \text{if } A_t \in \mathcal{S}, \tilde{E}_i(\theta) = E_i^* \\ 2 - \mathbf{1}\{\tilde{A}_{t,i} = E_i^*\} - 2 \cdot \mathbf{1}\{\tilde{A}_{t,i} = \tilde{E}_i(\theta))\} & \text{if } A_t \in \mathcal{S}, \tilde{E}_i(\theta) \neq E_i^* \\ \mathbf{1}\{\tilde{E}_i(\theta) = E_i^*\} & \text{if } A_t \in \mathcal{H} \end{cases} \right] \quad (17) \\
&\geq \sum_{t=1}^{n} \mathbb{E}_\theta \left[ \varepsilon \sum_{i=1}^{b} \begin{cases} 1 - \mathbf{1}\{\tilde{A}_{t,i} = E_i^*\} & \text{if } A_t \in \mathcal{S}, \tilde{E}_i(\theta) = E_i^* \\ \mathbf{1}\{\tilde{A}_{t,i} = E_i^*\} & \text{if } A_t \in \mathcal{S}, \tilde{E}_i(\theta) \neq E_i^* \\ 1 & \text{if } A_t \in \mathcal{H}, \tilde{E}_i(\theta) = E_i^* \\ 0 & \text{if } A_t \in \mathcal{H}, \tilde{E}_i(\theta) \neq E_i^* \end{cases} \right] \\
&= \mathbb{E}_\theta \left[ \varepsilon \sum_{i=1}^{b} \begin{cases} \sum_{t \in T_\mathcal{H}} 1 + \sum_{t \in T_\mathcal{S}} \left[ 1 - \mathbf{1}\{\tilde{A}_{t,i} = E_i^*\} \right] & \text{if } \tilde{E}_i(\theta) = E_i^* \\ \sum_{t \in T_\mathcal{S}} \mathbf{1}\{\tilde{A}_{t,i} = E_i^*\} & \text{if } \tilde{E}_i(\theta) \neq E_i^* \end{cases} \right] \\
&= \sum_{i=1}^{b} \varepsilon \cdot \begin{cases} \mathbb{E}_\theta \left[ n - \sum_{t \in T_\mathcal{S}} \mathbf{1}\{\tilde{A}_{t,i} = E_i^*\} \right] & \text{if } \tilde{E}_i(\theta) = E_i^* \\ \mathbb{E}_\theta \left[ \sum_{t \in T_\mathcal{S}} \mathbf{1}\{\tilde{A}_{t,i} = E_i^*\} \right] & \text{if } \tilde{E}_i(\theta) \neq E_i^* \end{cases}
\end{aligned}
$$

$$= \sum_{i=1}^{b} \frac{n\varepsilon}{2} \cdot \begin{cases} \mathbb{P}_\theta(D_i^c) & \text{if } \tilde{E}_i(\theta) = E_i^* \\ \mathbb{P}_\theta(D_i) & \text{if } \tilde{E}_i(\theta) \neq E_i^* \end{cases}$$

Recall that $T_{\mathcal{S}} := \{t \in [n] \,|\, A_t \in \mathcal{S}\}$ and $T_{\mathcal{H}} := \{t \in [n] \,|\, A_t \in \mathcal{H}\}$. In step (16), for the case $A_t \in \mathcal{H}$, we invoke the inequality in (15). Meanwhile, the case $A_t \in \mathcal{S}$ can be further analyised per extended-group as shown in step (17): when $A_t \in \mathcal{S}$, the optimal move for $A_t$ in each extended-group $i$ is to select the unit that $\theta$ selects, i.e. $\tilde{A}_{t,i} = \tilde{E}_i(\theta)$; doing so yields an optimal gain of $\varepsilon$ if $\tilde{E}_i(\theta) = E_i^*$ or an optimal gain of $2\varepsilon$ if $\tilde{E}_i(\theta) \neq E_i^*$.

## B.2 Proof of Claim 3.10

Given $i$ and $\tilde{\mathbf{E}}_{-i}$, we have

$$\sum_{E \in \mathcal{G}_i} \frac{n\varepsilon}{2} \cdot \left[ \mathbb{P}_{\theta(E_i^*, \tilde{\mathbf{E}}_{-i})}(\overline{D}_i) + \mathbb{P}_{\theta(E, \tilde{\mathbf{E}}_{-i})}(D_i) \right]$$

$$\geq \frac{n\varepsilon}{4} \sum_{E \in \mathcal{G}_i} \exp\left( -\mathrm{KL}(\mathbb{P}_{\theta(E_i^*, \tilde{\mathbf{E}}_{-i})}; \mathbb{P}_{\theta(E, \tilde{\mathbf{E}}_{-i})}) \right)$$

$$= \frac{n\varepsilon}{4} \sum_{E \in \mathcal{G}_i} \exp\left( -2\varepsilon^2 \mathbb{E}_{\theta(E_i^*, \tilde{\mathbf{E}}_{-i})} \left[ \sum_{t=1}^{n} \langle A_t, E \rangle^2 \right] \right)$$

$$\geq \frac{n\varepsilon|\mathcal{G}_i|}{4} \exp\left( -2\varepsilon^2 \mathbb{E}_{\theta(E_i^*, \tilde{\mathbf{E}}_{-i})} \left[ \sum_{t=1}^{n} \sum_{E \in \mathcal{G}_i} \frac{\langle A_t, E \rangle^2}{|\mathcal{G}_i|} \right] \right)$$

$$\geq \frac{n\varepsilon|\mathcal{G}_i|}{4} \exp\left( -2\varepsilon^2 \mathbb{E}_{\theta(E_i^*, \tilde{\mathbf{E}}_{-i})} \left[ \sum_{t \in T_{\mathcal{S}}} \max_{X \in \mathcal{S}} \Phi_{\mathcal{G}_i}(X) + \sum_{t \in T_{\mathcal{H}}} \max_{X \in \mathcal{H}} \Phi_{\mathcal{G}_i}(X) \right] \right)$$

$$\geq \frac{n\varepsilon|\mathcal{G}_i|}{4} \exp\left( -2\varepsilon^2 \mathbb{E}_{\theta(E_i^*, \tilde{\mathbf{E}}_{-i})} \left[ n \cdot \max_{X \in \mathcal{G}_i} \Phi_{\mathcal{G}_i}(X) + |T_{\mathcal{H}}| \cdot \max_{X \in \mathcal{H}} \Phi_{\mathcal{G}_i}(X) \right] \right)$$

$$\geq \frac{n\varepsilon|\mathcal{G}_i|}{4} \exp\left( -2\varepsilon^2 \mathbb{E}_{\theta(E_i^*, \tilde{\mathbf{E}}_{-i})} \left[ \frac{n}{p} + \frac{|T_{\mathcal{H}}|}{q} \right] \right)$$

$$\geq \frac{n\varepsilon|\mathcal{G}_i|}{4} \exp\left( -2\varepsilon^2 \left[ \frac{n}{p} + \frac{1}{q} \max_{\theta \in \Theta} \mathbb{E}_\theta[|T_{\mathcal{H}}|] \right] \right)$$

We consider

$$\mathbf{T} := ((E_1, E_1^*), (E_2, E_2^*), \dots, (E_b, E_b^*)) \in \bigtimes_{i=1}^{b} (\mathcal{G}_i \times \{E_i^*\})$$

Given $\mathbf{T}$, we denote

$$\tilde{\mathbf{E}} := (\tilde{E}_1, \tilde{E}_2, \dots, \tilde{E}_b) \in \bigtimes_{i=1}^{b} \{E_i, E_i^*\}$$

Thus, we have

$$\max_{\theta \in \Theta} R_n^\pi(\theta) \geq \frac{1}{\left|\bigtimes_{i=1}^b (\mathcal{G}_i \times \{E_i^*\})\right|} \sum_{\mathbf{T}} \frac{1}{\left|\bigtimes_{i=1}^b \{E_i, E_i^*\}\right|} \sum_{\tilde{\mathbf{E}}} R_n^\pi(\theta(\tilde{\mathbf{E}}))$$

$$\geq \frac{1}{\prod_{i=1}^b |\mathcal{G}_i|} \cdot \frac{1}{2^b} \sum_{\mathbf{T}} \sum_{\tilde{\mathbf{E}}} \sum_{i=1}^b \frac{n\varepsilon}{2} \cdot \begin{cases} \mathbb{P}_{\theta(\tilde{\mathbf{E}})}(\overline{D}_i) & \text{if } \tilde{E}_i = E_i^* \\ \mathbb{P}_{\theta(\tilde{\mathbf{E}})}(D_i) & \text{if } \tilde{E}_i \neq E_i^* \end{cases}$$

$$= \frac{1}{\prod_{i=1}^b |\mathcal{G}_i|} \cdot \frac{1}{2^b} \sum_{i=1}^b \sum_{\mathbf{T}_{-i}} \sum_{\tilde{\mathbf{E}}_{-i}} \sum_{E \in \mathcal{G}_i} \frac{n\varepsilon}{2} \cdot \left[ \mathbb{P}_{\theta(E_i^*, \tilde{\mathbf{E}}_{-i})}(\overline{D}_i) + \mathbb{P}_{\theta(E, \tilde{\mathbf{E}}_{-i})}(D_i) \right]$$

$$\geq \frac{1}{\prod_{i=1}^b |\mathcal{G}_i|} \cdot \frac{1}{2^b} \sum_{i=1}^b \sum_{\mathbf{T}_{-i}} \sum_{\tilde{\mathbf{E}}_{-i}} 2|\mathcal{G}_i| \cdot \frac{n\varepsilon}{8} \exp\left( -2\varepsilon^2 \left[ \frac{n}{p} + \frac{1}{q} \max_{\theta \in \Theta} \mathbb{E}_\theta[|T_\mathcal{H}|] \right] \right)$$

$$= \frac{nb\varepsilon}{8} \exp\left( -2\varepsilon^2 \left[ \frac{n}{p} + \frac{1}{q} \max_{\theta \in \Theta} \mathbb{E}_\theta[|T_\mathcal{H}|] \right] \right)$$

Then apply $3b \geq s$ to get the claimed lower bound (4).

