# OpenReview forum: "A Novel General Framework for Sharp Lower Bounds in Succinct Stochastic Bandits"
_NeurIPS.cc/2025/Conference — NeurIPS 2025 poster_

### Official Review · Reviewer_fbhz · 2025-06-29

**Clarity:** 3
**Significance:** 4
**Originality:** 3
**Rating:** 5
**Confidence:** 4

**Summary:**

This paper presents an algebraic approach to defining *succinctness* and constructs a framework for the linear bandit problem with a succinct parameter, which encompasses entry-sparse vectors, low-rank matrices, and group-sparse matrices. The paper provides a regret lower bound under this framework, and the resulting lower bounds for each application are either new or improved.

**Questions:**

1. In Corollary 4.2, it seems like it should be $Q(\cdot) = || \cdot ||_ {2, \infty}$ and $R(\cdot) = || \cdot ||_ {2, 1}$, which correspond to the maximum and the sum of $\ell_ 2$ norms of the rows, respectively. Could the authors confirm this, or clarify the exact definitions of the matrix norms in the paper?

2. I am curious about where the term *succinct* comes from. Is it a concept newly introduced in this paper, or is it an existing one? If it is the former, I suggest noting this in the abstract, as facing the word *succinct* before its description adds slight confusion.


**Minor comments (Typos)**

I could not find the definition of $\Delta_{\mathcal{X}}$, which appears for the first time in line 210. I could easily guess its definition, but I believe the authors would prefer to have it explicitly defined.
In line 224, should it not be $1/2$ instead of $1/4$?
In line 258, should it not be $C_{\min}(\mathcal{A}) > 0$ instead of $\ge 0$, as the latter is always true?

**Ethical Concerns:**

["NO or VERY MINOR ethics concerns only"]

**Final Justification:**

The paper provides a new and unified perspective on constructing lower bounds for three distinct problems, and the bounds improve upon the previous ones.
This paper has significant theoretical contributions and is technically sound.
The main concern that was raised by all the reviewers was lack of details and explantions in Sections 3 and 4, and the authors agreed that those sections are hard to follow and promised improvements.
Based on the premise that Sections 3 and 4 will contain more details in the final version as promised, I support the acceptance of the paper.

**Limitations:**

yes

**Paper Formatting Concerns:**

None.

**Quality:**

4

**Strengths And Weaknesses:**

Strengths

1. The paper identifies core and common properties of three distinct problems (entry-sparsity, low-rank, and group-sparsity) and conducts the analysis under minimal assumptions. This approach is indeed novel and general. I believe it sets a strong example of how theoretical approach can unify multiple problem classes.

2. The derived lower bounds improve upon previous ones, demonstrating the strength of the analysis.

3. The introduction of the $Q$ and $R$ norms in the framework provides insight into which norm should be used in the boundedness assumptions for individual problem settings.


Weaknesses

1. The final step of the proof of the main theorem, as well as the proofs of the corollaries, is somewhat difficult to follow. For instance, it took me some time to understand that the phrase “we obtain the lower bound $\min( \frac{ns\varepsilon}{24}\exp(-2\varepsilon [ \frac{n}{p}+ \frac{h}{q}]), \frac{C_0}{3}h )$” in line 237 means that the worst-case regret is lower bounded by this expression for *any* $h$, as the intermediate steps are largely omitted. In the proofs of the corollaries, I would appreciate a bit more elaboration on how the bounds for $C_{\min}(\mathcal{H})$, $p$, and $q$ are obtained. While I understand it is mostly tedious calculation, I believe it would be helpful for readers if the authors briefly indicated what exactly needs to be computed.

2. The final step of the proof of Theorem 3.8 introduces certain conditions on $n$, for example, $9s^2p \le C_0^2 n$ in line 240, which are not mentioned in the theorem statement. Including these assumptions explicitly would improve clarity.

3. The lower bound of $\Omega(\sqrt{sdn})$ from [11] is mentioned in the related work section but is omitted from Table 1, which could be slightly misleading.

---

> ### Author Rebuttal · Authors · 2025-07-31
>
> > 1. ... identifies core and common properties of three distinct problems... and conducts the analysis under minimal assumptions. This approach is indeed novel and general... sets a strong example of how theoretical approach can unify multiple problem classes.
>
> We sincerely thank the reviewer for the appreciation of our work.
>
> > 1. The final step of the proof of the main theorem, as well as the proofs of the corollaries, is somewhat difficult to follow. For instance, it took me some time to understand that the phrase "we obtain the lower bound $\min(...)$" in line 237 means that the worst-case regret is lower bounded by this expression for any h, as the intermediate steps are largely omitted. In the proofs of the corollaries, I would appreciate a bit more elaboration on how the bounds for $C_{\min}(\mathcal H)$, p, and q are obtained. While I understand it is mostly tedious calculation, I believe it would be helpful for readers if the authors briefly indicated what exactly needs to be computed.
>
> Yes. In line 237, we meant to say that by combining the lower bounds (4) and (5) in a minimum expression, we can replace the quantity $\max\_{\theta\in\Theta} \mathbb E \_\theta [|T_{\mathcal{H}}|]$ with a free variable $h$ (since one bound tightens by the growth of the constrained quantity and the other loosens) and then obtain the lower bound. We will rephrase it to stress that the bound $\min(...)$ holds for any $h$. We agree that the proofs of the corollaries would benefit from more elaboration. While the derivations involve mostly routine calculations --- especially so when one realizes that we are simply manipulating groups of basis vectors that can span the whole space and can form some succinct support --- we will revise both the main text and the appendix to provide clearer guidance.
>
> > 2. The final step of the proof of Theorem 3.8 introduces certain conditions on n, for example, $9s^2p \leq C_0^2n$ in line 240, which are not mentioned in the theorem statement. Including these assumptions explicitly would improve clarity.
>
> Thank you for the suggestion. As is common in the literature, lower bounds are typically stated in a "there exists an instance" manner to avoid overloading the theorem statement with all the construction details. Some minor construction conditions are often omitted, unless they are directly relevant to the compatibility with the assumption class of interest, such as data-poor and data-rich regimes, e.g., $d \ll n$. To fully verify compatibility, one must examine the specific instance constructed in the proof. For example, if there's some signal magnitude assumption imposed on $\theta$, one should also check whether the constructed $\theta$ satisfies it. That said, we agree that clarity is important, and we will consider discussing such conditions more explicitly or summarizing them in a remark.
>
> > 3. The lower bound of $\Omega(\sqrt{sdn})$ from [11] is mentioned in the related work section but is omitted from Table 1, which could be slightly misleading.
>
> You are right. This was an unintentional omission. We will update Table 1 to include it.
>
> > 1. In Corollary 4.2, it seems like it should be $Q(.) = \\| \cdot \\|\_{2,\infty}$ and $R(.) = \\| \cdot \\|\_{2,1}$, which correspond to the maximum and the sum of $\ell_2$ norms of the rows, respectively. Could the authors confirm this, or clarify the exact definitions of the matrix norms in the paper?
>
> Thanks for bringing it to our attention. We acknowledge that the notation for matrix norms in the paper may be unconventional and potentially ambiguous. To clarify, for a matrix $X\in \mathbb{R} ^{d_1\times d_2}$, we intended the following definitions:
>
> $\\| X \\|\_{1, 2} := \sum_{i=1} ^{d_1} \left(\sum_{j=1} ^{d_2} |x_{ij}|^2\right)^{1/2}$
>
> $\\| X \\|\_{\infty, 2} := \max_{1\leq i \leq d_1} \left(\sum_{j=1} ^{d_2} |x_{ij}|^2\right)^{1/2}$
>
> We will revise the notation to follow standard conventions, changing the subscript order, e.g. from $\\|.\\|\_{\infty, 2}$ to $\\|.\\|\_{2,\infty}$.
>
> > 2. I am curious about where the term succinct comes from. Is it a concept newly introduced in this paper, or is it an existing one? If it is the former, I suggest noting this in the abstract, as facing the word succinct before its description adds slight confusion.
>
> We developed the concept of succinctness in the context of identifying the common algebraic structure underlying these bandit problems. We chose the term "succinct" to distinguish it from "sparse", commonly seen in literature on the entry-sparse bandits. While the concept arose independently during our work and we did not find prior references specific to this setting, it is not hard at all to imagine that similar concepts have been explored, or even extensively studied in other domains or settings, as it represents a natural extension of general vector space theory. We appreciate the suggestion and will consider some further clarification in the abstract.
>
> > Minor comments (Typos)
>
> > I could not find the definition of $\Delta_\theta(X)$, which appears for the first time in line 210. I could easily guess its definition, but I believe the authors would prefer to have it explicitly defined.
>
> > In line 224, should it not be 1/2 instead of 1/4?
>
> > In line 258, should it not be $C_\min(\mathcal A)>0$ instead of $C_\min(\mathcal A) \geq 0$, as the latter is always true?
>
> Thank you for your careful review. Your comments are all valid. We clarify that $\Delta_\theta(X)$ represents the sub-optimal gap of any $X\in\mathcal{H}$ and is formally defined as
>
> $\Delta_\theta(X) := \max_{X^* \in \mathcal A} \langle X^*, \theta \rangle - \langle X, \theta\rangle$
>
> This will be explicitly introduced at its first appearance in the revised version. We also agree with the corrections in lines 224 and 258 and will update them accordingly.

---

> > ### Comment · Reviewer_fbhz · 2025-08-06
> >
> > I deeply appreciate the authors' detailed response to my comments.
> > I am satisfied with the authors' rebuttal and remain positive about the paper.
> > Still, the paper would greatly benefit by making it easier to follow.
> >
> > I would like to make one minor comment regarding the answer to the second weakness.
> > I interpreted the mentioned conditions as conditions on $n$. From this perspective,  authors' response seems slightly off-topic (although I fully understand and agree with it), since no lower-bound theorem would state “there exists $n$ such that...”.
> > Instead, it would be more rigorous to state the range of $n$ for which the lower bound holds (e.g., Theorems 15.2, 24.2 in [11]). I raised this point only because I had the impression that the authors care extra about technical rigor, and I am happy to hear that the authors will discuss the conditions more explicitly.

---

> > > ### Author Response · Authors · 2025-08-07
> > >
> > > Thank you very much for your follow-up on Weakness 2.
> > >
> > > The examples you mentioned, Theorems 15.2 and 24.2 in [11], indeed specify the valid range of $n$ with respect to $d$, as part of establishing their validity in the data-rich regime. In contrast, our result provides a unified lower bound, with different rates emerging in the data-poor and data-rich regimes.
> > >
> > > Nonetheless, we agree that it would improve rigor to explicitly discuss the conditions on $n$ in relation to other quantities beyond just $d$.  We will revise the text to include this specific discussion, along with additional guidance on certain derivations to improve overall readability.
> > >
> > > We appreciate your time and thoughtful feedback.

---

### Official Review · Reviewer_23qS · 2025-07-02

**Clarity:** 2
**Significance:** 4
**Originality:** 3
**Rating:** 5
**Confidence:** 4

**Summary:**

This paper proposes a novel “template” or framework to prove lower bounds on bandit problems with a so-called “succinct” structure (e.g. sparsity, defined in section 3). Following instantiations of their methods, the authors prove sharper or new lower bounds on Sparse bandits, Group-Sparse bandits and Low-Rank bandits.

**Questions:**

* I guess your framework applies to K-armed bandits as well since it’s a sub-case of the linear bandits. What are known lower bounds for the S-sparse K-armed bandits? Is your technique also improving that?
* Your results seems to highlight gaps between lower and upper bounds, either in the C_min dependency or in the other problem quantities. All sections 4.1, 4.2, 4.3 are purely technical and do not discuss the implications of your results. Can you comment on the gaps you opened and whether you believe the lower bounds could be attainable? or perhaps what may cause sub-optimality in the upper-bounds?

**Ethical Concerns:**

["NO or VERY MINOR ethics concerns only"]

**Final Justification:**

I have carefully read the responses as well as the other reviews. I maintain my opinion that this is a strong paper. My main suggestions for improvements are:
* Try to use a running example to showcase the method and give an intuition of how the construction actually works (especially in the smallest example where the method improves over standard arguments)
* Clarify some technical steps as discussed with reviewer fbhz (I did not explicitly mention these myself but I agree with the reviewer).

**Limitations:**

Overall, this is a strong theory paper that has many contributions and few limitations. Given that there is a bit more space on page 9, I would like the authors to discuss a bit the implications of their results (see questions).

**Quality:**

3

**Strengths And Weaknesses:**

### Strengths:

* The technical contribution is strong: the paper provides a clear and relatively novel method to prove lower bounds
* Results are significant and may have major impact on succinct bandits research because the lower bounds that are proved are mainly not matched by existing upper bounds

### Weaknesses:

* Perhaps some additional citations could be provided, in particular in section 2.2 (KL lower bound on l.112 “or other options”). For now this section gives the feeling that this general framework wasn’t known before in the bandit literature, though similar ideas are discussed in many papers and gathered nicely in Lattimore &Szepesvari (2020). In general, I would have liked to understand better what was done wrong before and that you fixed. Perhaps this is written somewhere, or understandable in between the line and I missed it but I felt like I kept asking myself “What is different from known frameworks”?
* Comments and discussions are lacking
* Some vocabulary in Section 3 is a bit confusing and not always easy to connect with the bandit problems. Generally Section 3 is the hardest section to follow because it tries to be as general as possible in order to instantiate results into applications afterwards. It makes sense as a presentation choice, but I find that I often prefer to anchor my thoughts into a concrete example so I found myself reading sections 4 and 3 back and forth and trying to connect the dots. I think it’s a minor comment and I will adjust my judgement according to other reviewers’ opinions. Perhaps a way to help the reader would be to use sparse linear bandits as a running example?

---

> ### Author Rebuttal · Authors · 2025-07-31
>
> > The technical contribution is strong: the paper provides a clear and relatively novel method to prove lower bounds
>
> > Results are significant and may have major impact on succinct bandits research because the lower bounds that are proved are mainly not matched by existing upper bounds
>
> > Overall, this is a strong theory paper that has many contributions and few limitations...
>
> We sincerely thank the reviewer for the appreciation of our work.
>
> > Perhaps some additional citations could be provided, in particular in section 2.2 (KL lower bound on l.112 "or other options"). For now this section gives the feeling that this general framework wasn't known before in the bandit literature, though similar ideas are discussed in many papers and gathered nicely in Lattimore &Szepesvari (2020)...
>
> Thank you for the comment. Section 2.2 (inside the "Preliminaries" section) is not part of our proposed framework, but rather serves as a brief overview of standard techniques and machinery used in the literature for lower bound construction --- tools that we also draw upon later in proving our main theorem. In hindsight, we agree that the lack of citation may have unintentionally given the impression that we were proposing new techniques in Section 2.2. We will revise the text to clarify its preparatory purpose and add appropriate references to acknowledge the origins and prior applications of these techniques. In particular, a citation to Lattimore & Szepesvári (2020) should have been included after "or other options" in line 112.
>
> > Comments and discussions are lacking
>
> > Some vocabulary in Section 3 is a bit confusing and not always easy to connect with the bandit problems. Generally Section 3 is the hardest section to follow because it tries to be as general as possible in order to instantiate results into applications afterwards. It makes sense as a presentation choice, but I find that I often prefer to anchor my thoughts into a concrete example so I found myself reading sections 4 and 3 back and forth and trying to connect the dots. I think it's a **minor comment and I will adjust my judgement according to other reviewers' opinions**. Perhaps a way to help the reader would be to use sparse linear bandits as a running example?
>
> We agree that Section 3 can be challenging to follow without concrete examples. Given the extra page for the final version, it is worthwhile to discuss between Section 3.1 and Section 3.2 how the three concrete cases, i.e. entry-sparse, group-sparse, and low-rank, fit into the general succinctness model. In particular, we will highlight what constitutes a succinct support in each case. For example, in the low-rank case, once we define the succinct unit set $\mathcal U$ to contain all the 1-rank matrices of unit length, then any non-empty subset of $\mathcal U$, e.g. $\\{u_iv_i^T\\}_{i=1}^{s}$ that fulfills $\langle{u_i},{u_j}\rangle=0$ and $\langle{v_i},{v_j}\rangle=0$ for distinct $i,j \in [s]$ constitutes a support. That way, when following the proof of the main theorem in Section 3.2, hopefully, the reader would have an intuitive interpretation of how a set of succinct units are selected from each group $\mathcal G_i$ to form a support. Originally, we had prepared three minor corollaries at the end of Section 3.1 to illustrate the three examples of the succinctness model, but we eventually compressed this material into the instantiations of general entities $\mathcal{U}$,$Q(\cdot)$ and $R(\cdot)$ at the beginning of the proof of the corollaries in Section 4.
>
> > I guess your framework applies to K-armed bandits as well since it's a sub-case of the linear bandits. What are known lower bounds for the S-sparse K-armed bandits? Is your technique also improving that?
>
> Yes, "s-sparse K-armed bandits" are essentially the entry-sparse case discussed in our paper, where $\theta \in \mathbb R^d$ has the number of non-zero entries limited to $s$. For canonical K-armed bandits where the agent aims to identify the arm with the highest expected reward, we can consider a sole non-zero-mean reward arm, that is to set $s=1$ and $d=K$.
>
> For the entry-sparse case. When the assumption class permits any arbitrary action set, [11] shows that $\Omega(\sqrt{sdn})$ can be constructed in the data-rich regime, i.e. assuming $d \ll n$, while $\Omega(n)$ can be trivially constructed in the data-poor regime. When the assumption class only considers those action sets that can admit a well-conditioned exploration distribution, i.e. $C_{\min}(\mathcal A) > 0$, [7] gives a lower bound of$\min(C_{\min}^{-1/3}s^{1/3}n^{2/3},\sqrt{dn})$, which is improved to $\min(C_{\min}^{-1/3}s^{2/3}n^{2/3},\sqrt{sdn})$in our Corollary 4.1.
>
> While it is a modest improvement when one constrains the scope to that specific assumption class, i.e. $C_{\min}(\mathcal A) > 0$ and entry-sparse $\theta$, we emphasize that our framework elevates the discussion by offering a unified view and enables extension of existing lower and upper bounds analysis in specific cases.
>
> > Your results seems to highlight gaps between lower and upper bounds, either in the C_min dependency or in the other problem quantities. All sections 4.1, 4.2, 4.3 are purely technical and do not discuss the implications of your results. Can you comment on the gaps you opened and whether you believe the lower bounds could be attainable? or perhaps what may cause sub-optimality in the upper-bounds?
>
> Essentially, our framework operates by manipulating succinct groups where some support can be constructed by selecting one element from each group. In the constructional proof, to obtain the final action set $\mathcal A := \mathcal S \cup \mathcal H$, we constructed $\mathcal S$-based actions that are s-succinct and thus uninformative when playing under the s-succinct parameters coming from the same succinct groups. Meanwhile, we impose that $\mathcal H$-based actions incur high regret under a specific instance $\theta_0$ coming from a "complementary succinct region". The trade-off yields our lower bound in the Theorem 3.8. In Section 4, we demonstrate that this general construction is valid when instantiated to the three specific assumption classes, i.e.$C_{\min}(\mathcal A) > 0$ in the three cases.
>
> For the entry-sparse case, the resulting lower bound is optimal in the data-rich regime and nearly optimal by a gap of $C_{\min}^{-1/3}$ in the data-poor regime, since there are algorithms offering matching upper bounds. From corollaries in Section 4, we believe that the construction framework can yield rather tight lower bounds in a generalized $C_{\min} > 0$ assumption class. Proving so would likely require generalization of those algorithms, which is beyond the scope of our current study. However, from an intuitive standpoint, our framework lower-bounds the regret by imposing that $\mathcal H$-based actions incur constant regret, and by optimistically assuming that $\mathcal H$-based actions could be informative with respect to those s-succinct parameters, unlike s-succinct $\mathcal S$-based actions. That optimistic assumption is materialized in the corollaries by imposing $C_{\min}(\mathcal H) > 0$.
>
> We also recognize that there may be alternative ways to construct this information-regret trade-off. In its current form, the framework simply uses Assumption 3.7 (1), i.e. the constant sub-optimal gap under certain $\theta_0$, to discourage playing the potentially informative $\mathcal H$-based actions. This prevents degenerate cases such as a wildcard action $(1,1,1,...,1)$ that could always be played with zero regret under any s-sparse $\theta\in \mathbb R^d$ with positive entries. It would be interesting to investigate possible replacements of the Assumption 3.7 (1) in our framework.

---

> > ### Comment · Reviewer_23qS · 2025-08-02
> > **Acknowledgement and comment**
> >
> > Dear authors,
> >
> > I have read all the reviews and your rebuttals. It seems to me that most of us actually struggled to understand your method and I think this is an issue that you only partially address. We all do challenging things on hard topics, and conveying your ideas through clear writing is your duty. So just to respond to your rebuttal to reviewer aWds, yes it is expected that you would be penalized by that if all of us are confused. I think I am rather familiar with this type of proof techniques and I did feel like it was difficult to follow your contributions. I am not sure I follow your rebuttal much better.
> >
> > I am really confused by your comment on the K-armed bandits:
> > > Yes, "s-sparse K-armed bandits" are essentially the entry-sparse case discussed in our paper, where $\theta \in \mathbb R^d$ has the number of non-zero entries limited to $s$. For canonical K-armed bandits where the agent aims to identify the arm with the highest expected reward, we can consider a sole non-zero-mean reward arm, that is to set $s=1$ and $d=K$.
> >
> > I disagree in many ways with what's written above. First, the s-sparse K-armed bandit is only equivalent to "$\theta \in \mathbb R^d$ has the number of non-zero entries limited to $s$" if you impose the action set of be the canonical basis with $d=K$.
> >
> > Then, this is not really accurate as in the regret minimization setting we do not seek to identify the best arm strictly speaking:
> > > For canonical K-armed bandits where the agent aims to identify the arm with the highest expected reward
> >
> > But most importantly, I confused by this part:
> > > sole non-zero-mean reward arm, that is to set $s=1$ and $d=K$
> > What do you mean?? No, the s-sparse K-armed bandit is when there are s non-zero arms with one arm having a higher mean than the others (in the simpler case where there's a single best arm).
> >
> > I brought up s-sparse K-armed bandits because I think they often serve as simpler illustrative example to understand a potentially more general idea. But now I'm confused by your answer to this question. Can you please try to explain to me again how do you get this improvement in Corollary 4.1 in the K-armed s-sparse bandits?

---

> > > ### Author Response · Authors · 2025-08-04
> > >
> > > Dear Reviewer,
> > >
> > > Thank you for the comment, and we apologize for the confusion. Due to the *lack of references* in the reviewer's original question, we interpreted the term "s-sparse K-armed bandits" within the context of our work and misunderstood the reviewer's intended meaning.
> > >
> > > Regarding our *rebuttal's comment* "For canonical K-armed bandits where the agent aims to identify the arm with the highest expected reward" we acknowledge this was incorrect. Our intended meaning was limited to the illustrative case with a single non-zero (more precisely, positive)-mean arm, and was expressed informally.
> > >
> > > *To clarify, our manuscript does not discuss K-armed bandits or best-arm identification. We focus on how the shape of action set affects regret minimization and have provided a simple illustrative example in $\mathbb R^d$, which are entry-sparse bandit, also known as Lasso bandit or sparse linear bandit (see Table 1).*
> > >
> > > Given your latest comment, our current understanding is that our contributions may not have implications for the "s-sparse K-armed bandits" setting. If the action set is fixed to be the standard basis, consisting of 1-succinct actions, then this assumption class simply excludes our constructed instances with $s$-succinct actions in their final action set.
> > >
> > > It is possible that our current understanding is still not fully aligned. In that case, we would appreciate if you could point us to *formal definitions and theorems or specific papers* to which we should relate. We believe this would help make the discussion more focused and precise.
> > >
> > > Thank you for your time.

---

> > > > ### Comment · Reviewer_23qS · 2025-08-06
> > > > **Ok agreed**
> > > >
> > > > Ok, I was confused by your first response, but now I actually think I'm much more aligned with what you wrote and that confirms my initial good impression of this work.
> > > >
> > > > I think all of us had a hard time understanding some parts of your technical arguments. The reviews raise more or less detailed points which I think should be addressed. However, I will maintain my score.

---

### Official Review · Reviewer_aWds · 2025-07-03

**Clarity:** 2
**Significance:** 2
**Originality:** 3
**Rating:** 4
**Confidence:** 1

**Summary:**

A new model of "succinct" bandits is introduced that generalizes some previously studied models. Their model of succinctness relaxes the notion of orthonormal sets in a normed vector space. This allows the lower bound to apply to a variety of settings. For instance, when the 1-succinct points are 1-sparse vectors, s-succinct vectors correspond to k-sparse vectors, and so on. The authors apply their technique to a few settings establishing the first lower bounds for low rank and "group sparse" matrix bandits, and improving the lower bound on entry-sparse bandits.

I think of this as similar in spirit to the theory of matroids for greedy algorithms, where a specific structure in the instance guarantees approximation guarantees.

**Questions:**

1. A direct comparison between succinctness and sparsity would be useful in Section 3.1. It seems like succinctness itself doesnt imply much structure and is just a set that could be specified in a problem, while the notion of a support is similar to that of orthonormality/independence, and s-succinct vectors _do_ have implied some structure. I think this is confusing, but there must be some reason for not just writing s-succinct vectors as decomposing into 1-succinct vectors, but rather as decomposing into a support of 1-succinct vectors.
2. What is the purpose of the first part of Assumption 3.7 (i)? The lower bound just needs the existence of an action space, so I dont see why you need to impose a restriction on it as an assumption (that is, why do you need to say that $C_0 < 0$? If $C_0>0$ for some action set, then maybe the lower bound is just not valid).
3. What do these assumptions mean for the entry sparse bandit? Specifically, does Assumption 3.7 say that the ground truth $\theta$ is $k$-succinct? If so, what is the significance of the parameter $s$? How does Theorem 3.8 improve on the entry sparse lower bound? Is it just a better lower bound instance (so there is now a proof for the improved lower bound purely in terms of sparsity)? While this is not a necessary component of the narrative, it would be helpful in appreciating the technical merit of the paper.

**Ethical Concerns:**

["NO or VERY MINOR ethics concerns only"]

**Final Justification:**

Thank you for your response. I will keep my score.

**Limitations:**

Yes

**Quality:**

3

**Strengths And Weaknesses:**

Strengths:
The authors provide a novel generalization that yields to lower bounds for bandit problems that have additional "sparsity" structure. This seems like a fruitful generalization, and beside the possibility of resulting in further lower bounds in other settings, it seems to extract the notion of sparsity relevant for bandit algorithms.

Weakness:
The paper is very dense.
1. Proving that the instances in Section 4 actually satisfy Assumption 3.7. For instance, line 273, 288 just say "We can verify that ...".
2. Explaining in terms of sparsity what Assumptions 3.7 (ii) and (iii) mean. It seems like (ii) imposes a restriction on what is allowed to be denoted "succinct" vectors. Is $\max \Phi$ related to $C_min$ used later? If we just take $p=q=1$, then we already get a lower bound that differs mostly in $s$ terms, so is this specific useful in distinguishing this work from prior work?
3. Where does the construction in line 203 of $\Theta$ come from? Again I think that a concrete example with sparsity would make this easier to read.

Overall, I think this paper could be a little longer, with some proofs explained in more detail. The key to using this framework elsewhere is understanding Assumption 3.7, and I dont think it is clear how to even verify these assumptions.

---

> ### Author Rebuttal · Authors · 2025-07-31
>
> > ... similar in spirit to the theory of matroids for greedy algorithms, where a specific structure in the instance guarantees approximation guarantees.
>
> > ... provide a novel generalization that yields to lower bounds for bandit problems that have additional "sparsity" structure... fruitful generalization, and beside the possibility of resulting in further lower bounds in other settings, it seems to extract the notion of sparsity relevant for bandit algorithms.
>
> We sincerely thank the reviewer for the appreciation of our work.
>
> > The paper is very dense.
>
> We believe we did the best one can do given the complexity and challenge of our topic. We sincerely hope we are not penalized due to this.
>
> > 1. Proving that the instances in Section 4 actually satisfy Assumption 3.7...
>
> > 2. ... It seems like (ii) imposes a restriction on what is allowed to be denoted "succinct" vectors. Is $\max \Phi$ related to $C_{\min}$ used later? If we just take $p=q=1$, then we already get a lower bound that differs mostly in s terms, so is this specific useful in distinguishing this work from prior work?
>
> > 3. Where does the construction in line 203 of $\Theta$ come from?...
>
> Thank you for the comment. Understanding Assumption 3.7 is key to applying our framework to other problem classes. We acknowledge that, as currently stated in terms of general entities, the assumptions can be difficult to interpret on a first read. To provide an intuitive interpretation: Assumption 3.7 essentially says that if the assumption class permits some bandit instance with an action set $\mathcal H$, that performs universally poorly under a specific $\theta_0$ whose support lies in a certain "succinct region", then we can construct an additional set of s-succint actions $\mathcal S$ from "complementary" succinct groups, and include them in the final action set $\mathcal A:= \mathcal H \cup \mathcal S$ (provided this is allowed by the assumption class). More Informally, Assumption 3.7 and Theorem 3.8 together tell the story that if we can have a bandit instance with a specific $\mathcal H$ action set, that is in some sense geometrically flawed, then we can inject succinct/sparse actions to make the instance harder.
>
> $\max \Phi$ is not directly related to the generalized $C_{\min}$ assumption class introduced in Section 4. Instead, it arises naturally from the geometric interplay between the $\mathcal{H}$ action set and the succinct groups $\mathcal{G}\_i$, influencing the tightness of the resulting lower bound. That said, we believe there may be deeper connections between $\max \Phi$ and $C_{\min}$ that help explain why our construction yields nearly tight bounds under the generalized $C_{\min}$ assumption class.
>
> You are right that $p = q = 1$ always holds, since $\Phi_{\mathcal{G}\_i}$ can be interpreted as an "averaged squared projection" onto $\mathcal{G}$. However, these parameters still influence the tightness of the final lower bound. As shown in Section 4, under the generalized $C_{\min}$ assumption class, the difference primarily appears as a missing factor of $C_{\min}^{-1/3}$ in the data-poor regime and a $d_2 / d_1$ factor in the data-rich regime. We note that for the low-rank and group-sparse cases, the magnitudes of $C_{\min}^{-1/3}$ and $d_2 / d_1$ are comparable.
>
> The construction of $\Theta$ in line 203 is inspired by the one in Section 15.2 of Lattimore & Szepesvári (2020), where you can see how the sole $(\Delta, 0,0,...,0)$ and variants of $(\Delta, 0,...,2\Delta,..., 0)$ correspond to the different succinct unit selection $\tilde {E}_i \in \mathcal{G_i} \cup \{ E^*_i\}$.
>
> We now provide more detailed instructions for verifying Assumption 3.7 in the entry-sparse case. We define $\mathcal U := \{e_j \}^d_{j=1}$ as the standard basis of $\mathbb{R}^d$, so any non-empty subset of $\mathcal{U}$ forms a succinct support. Meanwhile, we can define
>
> $\mathcal H := \{(1, \pm\kappa, \pm\kappa,..., \pm\kappa, \pm\kappa)^T\} \subset \mathbb R^d$
>
> where each $\mathcal{H}$-based action has its first entry fixed at 1 and the remaining entries equal to $\kappa$ or $-\kappa$. We can see that $C_{\min}(\mathcal H) \ge \kappa^2$ as the minimum eigenvalue of the population covariance matrix from uniformly sampling actions from $\mathcal H$ will be $\kappa^2$. Playing any $\mathcal H$-based action under a specific $\theta_0:= -e_1 =(-1, 0,0,...,0,0)^T$ will incur a constant regret of $1$ compared to some action whose first entry is 0.
>
> Now, we define $s-1$ groups where we can form a support of cardinality $s-1$ by selecting one element from each group. Importantly, the support should be "succinct complementary" with a support of $\theta_0$ , e.g. $\{e_1\}$. Take $d= 12$ and $s=3$ for example. We can define
>
> $\mathcal G_1 =\\{e_5, e_6, e_7, e_8\\}, \quad \mathcal G_2 =\\{e_9, e_{10}, e_{11}, e_{12}\\}$
>
> It is easy to see that since elements in $\mathcal G_i$ are mutually orthogonal, for any $x\in \mathcal G_i$
>
> $\Phi_{\mathcal G_i}(x) := \sum_{e\in \mathcal G_i} \langle x, e\rangle^2/|\mathcal G_i| = 1/|\mathcal G_i| = 1/ (d/s)$
>
> It is also easy to see that for any $x\in \mathcal H$, the above quantity will be upper bounded by $\kappa^2$, since for each $e \in \mathcal G_i$ we have $\langle e, x\rangle^2 = \kappa^2$.
>
> > 1. A direct comparison between succinctness and sparsity would be useful in Section 3.1. It seems like succinctness itself doesnt imply much structure and is just a set that could be specified in a problem, while the notion of a support is similar to that of orthonormality/independence, and s-succinct vectors do have implied some structure...
>
> You are right that succinctness itself doesn't imply much structure, in the sense that we can just define a different succinct unit set $\mathcal U$. These succinct units are just pre-defined to be "1-succinct".
>
> An s-succinct vector is eventually defined via its decomposition to a support of s cardinality. A support is a set of succinct units that collectively uphold the relationship given in Eq (2), with respect to themselves and all the succinct units in $\mathcal U$. That ensures that the whole or any part won't collapse into just "1-succinct". For example, adding $s$ 1-sparse vectors does not necessarily yield an $s$-sparse vector, and similarly, the sum of $s$ rank-1 matrices may not have rank $s$.
>
> Our succinctness model subsume the definition of sparsity in entry-sparse case, where an s-sparse vector could alternatively be defined as decomposable to a disjointed subset of the standard basis or just the $\mathcal U$. The is a special case of succinctness, where the $\mathcal U$ only contains mutually orthogonal units. When $\mathcal{U}$ is the set of all rank-1 matrices of unit length, for instance, the units are no longer mutually orthogonal, and so a more general way is needed to define decomposition and support.
>
> > 2. What is the purpose of the first part of Assumption 3.7 (i)? The lower bound just needs the existence of an action space...
>
> Our framework establishes lower bounds by constructing hard bandit instances within a given assumption class. A bandit instance determines a stochastic process where a concrete regret can be computed, meaning it specifies how action options and rewards are generated. In our final constructed bandit instance, the action set $\mathcal A := \mathcal S \cup \mathcal H$.
>
> Assumption 3.7 (ii) enables the construction of $\mathcal S$-based actions, which are s-succinct and thus uninformative when playing under s-succinct parameters coming from the same succinct groups. Meanwhile in Assumption 3.7 (i) ensures that $\mathcal H$-based actions incur a constant regret under a specific instance $\theta_0$ coming from a "complementary succinct region", discouraging the playing of $\mathcal H$-based actions.
>
> We encapsulate these essential construction requirements in Assumption 3.7, so that one can assess whether our framework is compatible with the assumption class relevant to their setting. As an extreme counterexample, one can consider the assumption class where an wildcard action $(1,1,1,...,1) \in \mathbb R^d$ is always available in the action set and where the s-sparse $\theta$ only has positive entries. Our construction would be not valid in such an assumption class. A better phrasing in Assumption 3.7 (i) might be "the assumption class permits the inclusion of such an action set $\mathcal H$", rather than "there exists an action set $\mathcal H$"
>
> > 3. ... does Assumption 3.7 say that the ground truth $\theta$ is k-succinct? If so, what is the significance of the parameter s? How does Theorem 3.8 improve on the entry sparse lower bound?... While this is not a necessary component of the narrative, it would be helpful in appreciating the technical merit of the paper.
>
> No, Assumption 3.7 (i) does not say that the ground truth $\theta$ is k-succinct. All it says is that $\mathcal H$-based actions will get a reward with a negative mean that's small than $-C_0$ when playing under a specific instance $\theta_0$ that has a support of cardinality $k$. In Corollary 4.1, we eventually pick $\theta_0 := (-1, 0,0,...,0)$ with a support of cardinality $k=1$.
>
> For the entry-sparse case. When the assumption class only considers those action sets that can admit a well-conditioned exploration distribution, i.e. $C_{\min}(\mathcal A) > 0$, (7) gives a lower bound of $\min(C_{\min}^{-1/3}s^{1/3}n^{2/3},\sqrt{dn})$, which is improved to $\min(C_{\min}^{-1/3}s^{2/3}n^{2/3},\sqrt{sdn})$ in our Corollary 4.1.
>
> While it is a modest improvement when one constrains the scope to that specific assumption class, i.e. $C_{\min}(\mathcal A) > 0$ and entry-sparse $\theta$, we highlight that our framework elevates the discussion by offering a unified view and enables extension of existing lower and upper bounds analysis in specific cases.

---

> > ### Comment · Reviewer_aWds · 2025-08-09
> >
> > Thank you for your response. I believe all of my questions have been addressed.

---

### Official Review · Reviewer_hinF · 2025-07-03

**Clarity:** 3
**Significance:** 2
**Originality:** 2
**Rating:** 4
**Confidence:** 3

**Summary:**

This paper proposes a general framework for deriving regret lower bounds in linear bandit problems where the loss vectors exhibit some form of structural simplicity. Concrete examples of such structure include sparse vectors, low-rank matrices, and group-sparse matrices. The framework aims to encompass these cases in a unified manner. In particular, for the sparse linear bandit setting, the paper derives an improved lower bound compared to existing results.

**Questions:**

Regarding Theorem 3.8, can the authors provide concrete examples where the bound becomes tight (or provably loose)? Would it be possible to state sufficient conditions under which the lower bound is tight?

**Ethical Concerns:**

["NO or VERY MINOR ethics concerns only"]

**Final Justification:**

All of my questions have been addressed, and I have no remaining concerns. I have also confirmed that the concerns raised by other reviewers have been addressed. While I would not consider the paper to have particularly high impact, it does not have any major weaknesses either. I would therefore like to maintain my current borderline accept rating.

**Limitations:**

Yes

**Quality:**

3

**Strengths And Weaknesses:**

Strengths:

The problem is formulated with technical precision, and the theoretical results are well structured.

The presentation emphasizes generality and extensibility, suggesting that the framework could potentially be applied to other problem settings.

Weaknesses:

The intuitive explanation of the model definitions and technical results is somewhat limited. For example, in Section 3.1, while several notations and definitions are introduced, they are mostly listed without accompanying interpretations or illustrative examples. Providing such context would improve readability.

It would be helpful to include a more detailed discussion of how tight the proposed lower bounds are in general. For instance, regarding Theorem 3.8, can the authors provide concrete examples where the bound becomes tight (or provably loose)? Would it be possible to state sufficient conditions under which the lower bound is tight?

---

> ### Author Rebuttal · Authors · 2025-07-31
>
> > The problem is formulated with technical precision, and the theoretical results are well structured.
>
> > The presentation emphasizes generality and extensibility, suggesting that the framework could potentially be applied to other problem settings.
>
> We sincerely thank the reviewer for the appreciation of our work.
>
> > The intuitive explanation of the model definitions and technical results is somewhat limited. For example, in Section 3.1, while several notations and definitions are introduced, they are mostly listed without accompanying interpretations or illustrative examples...
>
> We have realized, from your comment and similar feedback from other reviewers, that Section 3 may be hard to follow without concrete examples. In response, we plan to add a short discussion between Sections 3.1 and 3.2 to illustrate how the three motivating cases --- entry-sparse, low-rank, and group-sparse --- fit into our succinctness model. Specifically, we will highlight the forms a succinct support takes in those three examples, and the instantiations of $Q(\cdot)$ and $R(\cdot)$. Right now, this discussion is compressed and deferred until the corollaries in Section 4. By moving it earlier, we hope to help readers anchor the abstract definitions in familiar examples --- particularly before Section 3.2 introduces Assumption 3.7 and Theorem 3.8, which work with general entities.
>
> For instance, Assumption 3.7(ii) requires selecting one element from each group to form a support. In the entry-sparse case, this corresponds to dividing the standard basis into groups, since any non-empty subset of the basis forms a support. Similarly, for the entry-sparse case, Assumption 3.7(i) essentially means that the specific instance $\theta_0$ is supported on the k standard basis vectors that are disjoint from those used by the groups. We hope the earlier inclusion of examples improves accessibility of the definitions and results, and would appreciate your thoughts on that.
>
> > It would be helpful to include a more detailed discussion of how tight the proposed lower bounds are in general.
>
> > Regarding Theorem 3.8, can the authors provide concrete examples where the bound becomes tight (or provably loose)? Would it be possible to state sufficient conditions under which the lower bound is tight?
>
> Thank you for the thoughtful question. Intuitively, the lower bound in Theorem 3.8 is established by constructing an instance where $\mathcal H$-based actions incur high regret under certain $\theta_0$ but could "potentially" be informative when playing under those s-succinct parameters formed from the succinct groups, unlike those $\mathcal S$-based actions. We can image that the bound becomes loose when this optimistic assumption is violated in some instances in the assumption class. When $\mathcal H$-based actions are as uninformative as sepearately pulling d arms in n rounds, the regret can be as large as $\Omega(n)$ in the data-poor regime, which is tighter than our bound of $\Omega(n^{2/3})$.
>
> Concretely, if the assumption class of interest includes instances with only those s-succinct actions in $\mathcal S$, that is to assume $\mathcal H = \emptyset$, Our construction will result in a lower bound of $s\sqrt{pn}$. For the entry-sparse case where $\mathcal S$ is constructed from the groups that divide the standard basis of cardinality $d$, it can eventually leads to $\Omega(\sqrt{sdn})$ in the data-rich regime which is tight due to an existing algorithm with matching upper bound that works on arbitrary action set, and to $\Omega(n)$ in the data-poor regime. We tend to believe that the resulting general lower bound of $s\sqrt{pn}$ in the data-rich regime is somewhat tight for such an assumption class, given the entry-sparse case.
>
> On the other hand, in Section 4, when the construction is applied to a general $C_{\min}$ assumption class in the three cases, we can successfully construct an $\mathcal H$ such that it both (i) has a constant regret gap and (ii) satisfies $C_{\min}(\mathcal H) > 0$. The latter condition means that sampling $\mathcal H$-based actions can give us well-conditioned exploration, i.e. they are indeed "informative". Again, for the entry-sparse case, the lower bound in Theorem 3.8 yields a lower bound $\Omega(C_{\min}^{-1/3}s^{2/3}n)$ in the data-poor regime, which is tight up to a $C_{\min}^{1/3}$ factor since an existing algorithm guarantees a matching upper bound. This gives us some confidence that the resulting lower bounds for the other two cases, or even for such a general $C_{min}$ assumption class beyond the three cases, is somewhat tight in the data-poor regime.
>
> We currently can only offer informal arguments for tightness, as formally proving tight lower bounds typically requires designing matching upper-bound algorithms, which lies beyond the scope of this work. Nonetheless, we believe the open questions raised by our work provide promising directions and motivation for extending existing lower-bound analysis and developing new algorithms.

---

> > ### Comment · Reviewer_hinF · 2025-08-05
> >
> > Thank you very much for your detailed response. All of my questions have been addressed, and I am fully satisfied with the explanations. I have no further questions.

---

### Decision · Program_Chairs · 2025-09-17

**Decision:**

Accept (poster)

**Comment:**

This paper studies lower bounds for linear bandits under structural assumptions like sparsity of low-rankedness. The reviewers agree the results are correct and constitute a significant contribution to the mature area of linear bandits. While some reviewers brought up concerns about tightness and required clarifications, these issues were resolved during the review process.